# Improving cell-free glycoprotein synthesis by characterizing and enriching native membrane vesicles

Jasmine M. Hershewe [1,2,3,12], Katherine F. Warfel [1,2,3,12], Shaelyn M. Iyer[1], Justin A. Peruzzi [1,2,3], Claretta J. Sullivan[4], Eric W. Roth[5], Matthew P. DeLisa[6,7,8], Neha P. Kamat[2,3,9] & Michael C. Jewett [1,2,3,10,11 ✉]

Cell-free gene expression (CFE) systems from crude cellular extracts have attracted much attention for biomanufacturing and synthetic biology. However, activating membrane-dependent functionality of cell-derived vesicles in bacterial CFE systems has been limited. Here, we address this limitation by characterizing native membrane vesicles in *Escherichia coli*-based CFE extracts and describing methods to enrich vesicles with heterologous, membrane-bound machinery. As a model, we focus on bacterial glycoengineering. We first use multiple, orthogonal techniques to characterize vesicles and show how extract processing methods can be used to increase concentrations of membrane vesicles in CFE systems. Then, we show that extracts enriched in vesicle number also display enhanced concentrations of heterologous membrane protein cargo. Finally, we apply our methods to enrich membrane-bound oligosaccharyltransferases and lipid-linked oligosaccharides for improving cell-free *N*-linked and *O*-linked glycoprotein synthesis. We anticipate that these methods will facilitate on-demand glycoprotein production and enable new CFE systems with membrane-associated activities.

[1] Department of Chemical and Biological Engineering, Northwestern University, Technological Institute E136, Evanston, IL 60208, USA. [2] Chemistry of Life Processes Institute, Northwestern University, Evanston, IL 60208, USA. [3] Center for Synthetic Biology, Northwestern University, Technological Institute E136, Evanston, IL 60208, USA. [4] Air Force Research Laboratory, Materials and Manufacturing Directorate, Wright-Patterson Air Force Base, Dayton, OH 45433, USA. [5] Northwestern University Atomic and Nanoscale Characterization and Experimentation (NUANCE) Center, Tech Institute A/B Wing A173, Evanston, IL 60208, USA. [6] Robert F. Smith School of Chemical and Biomolecular Engineering, Cornell University, Ithaca, NY 14853, USA. [7] Nancy E. and Peter C. Meinig School of Biomedical Engineering, Cornell University, Ithaca, NY 14853, USA. [8] Biomedical and Biological Sciences, College of Veterinary Medicine, Cornell University, Ithaca, NY 14853, USA. [9] Department of Biomedical Engineering, Northwestern University, Technological Institute E310, Evanston, IL 60208, USA. [10] Robert H. Lurie Comprehensive Cancer Center, Northwestern University, Chicago, IL 60611, USA. [11] Simpson Querrey Institute, Northwestern University, Chicago, IL 60611, USA. [12] These authors contributed equally: Jasmine M. Hershewe, Katherine F. Warfel. ✉email: m-jewett@northwestern.edu

Cell-free gene expression (CFE) systems activate transcription and translation using crude cellular extracts instead of living, intact cells[1]. In recent years, these systems have matured from widely used tools in molecular biology to platforms for biomanufacturing and synthetic biology[1–4]. Among CFE systems, *Escherichia coli*-based methods have been used the most[1,5–10]. A body of work dedicated to optimization of extract preparation and reaction conditions has simplified, expedited, and improved the cost and performance of *E. coli* CFE systems[1,11–13]. Optimized *E. coli*-based CFE reactions: (i) quickly synthesize grams of protein per liter in batch reactions[7,14–17]; (ii) are scalable from the nL to 100 L scale[18,19]; and (iii) can be freeze-dried for months of shelf stability[1,12,20–26]. The ability to readily store, distribute, and activate freeze-dried cell-free systems by simply adding water has opened new opportunities for point-of-use biosensing[27–33], portable therapeutic and vaccine production[22,23,34], and educational kits[1,35–37]. Thus, efforts to improve the efficiency and expand capabilities of engineered CFE systems could have impacts across many disciplines.

In a growing number of contexts, CFE extracts have been tailored to new applications by pre-enriching soluble, heterologous components in vivo prior to cell lysis, avoiding the need to separately purify and add these components. Examples where cells for CFE systems have been engineered prior to generating extract include incorporation of site-specific non-canonical amino acids into proteins[15,38,39], biosensing of analytes[31,40,41], and assembly of metabolic pathways for production of small molecules[9,42–44]. Heterologous components can also be directly added to the CFE system to achieve a desired function. For example, membrane augmented cell-free systems achieved through the use of nanodiscs, synthetic phospholipid structures, purified microsomes, and purified vesicles have all enabled the use of membrane-associated components in CFE systems[45–51].

Enabling membrane-associated functions in CFE systems is important for numerous biological functions. Indeed, lipid membranes play pivotal roles across all domains of life, with ~20–30% of genes encoding for membrane proteins and many essential processes taking place on and across membranes[52,53]. For example, membranes are required for molecular transport, immunological defense, energy regeneration, and post-translational protein modification (e.g., glycosylation).

Despite the absence of intact cellular membranes, membrane structures are present in crude extract-based CFE systems. They form upon fragmentation and rearrangement of cell membranes during cell lysis and extract preparation and have been studied and characterized for decades[51,54–57]. For example, oxidative phosphorylation and protein translocation were originally studied from purified vesicles prepared from *E. coli* cell extracts[58]. In *E. coli* CFE systems, inverted membrane vesicles harboring electron transport chain machinery activate oxidative phosphorylation and ATP regeneration[54,59]. Analogously, in eukaryotic-derived crude extract-based CFE systems, endoplasmic reticulum (ER)-derived microsomes enhance functionality, enabling the synthesis of membrane proteins and proteins with disulfide bonds, among others[5,26,60–65].

While there are some examples of cell-derived membrane-incorporated components to enhance bacterial CFE systems, this area of research has remained under-studied (with membrane augmented systems like nanodiscs being used most frequently). Yet, enriching native membrane-bound components in CFE systems, especially with heterologously expressed cargo, is poised to enable compelling applications. For example, protein glycosylation, which can profoundly impact folding, stability, and activity of proteins and therapeutics[66–69], is mediated by membrane-bound components. Introduction of cell-derived vesicles with machinery required for glycosylation could enable cell-free biomanufacturing of protein therapeutics and conjugate vaccines, especially at the point-of-need.

Along these lines, we recently described cell-free glycoprotein synthesis (CFGpS), a platform for one-pot biomanufacturing of defined glycoproteins in extracts enriched with heterologous, membrane-bound glycosylation machinery[34]. To date, CFGpS has been used to produce model glycoproteins, human glycoproteins, and protective conjugate vaccines[23,34,70–72]. Unfortunately, the existing CFGpS system based on S30 extracts (i.e., cell extracts that result from a 30,000g clarification spin) is limited by glycosylation efficiency, only producing ~10–20 µg/mL of glycoprotein in batch[2,34]. Characterizing and enriching cell-derived vesicles comprising membrane-bound glycosylation components offers one strategy to address this limitation, and perhaps make possible a variety of applications involving membrane-bound biology.

Here, we set out to develop methods that enhance membrane-dependent functionality of cell-derived vesicles in bacterial CFE systems, with a focus on bacterial glycoengineering. First, we characterize size distributions and concentrations of native membrane vesicles in extracts, providing a benchmark for analysis and engineering of CFE systems. To do so, we apply canonical strategies (e.g., TEM), and also apply simple and expedited characterization workflows that rely on techniques such as light scattering to directly analyze vesicles in extracts without the need for lengthy protocols. Second, we investigate the impacts of upstream extract processing steps on vesicle profiles, revealing simple handles to modulate vesicle concentration in extracts. Third, we use cell-derived membrane vesicles to enrich a variety of heterologous, membrane-bound proteins and substrates in extracts without the use of synthetically derived membranes. Finally, we apply our findings to improve glycoprotein yields in our existing asparagine-linked (*N*-linked) CFGpS system and a new membrane-dependent CFGpS system based on serine/threonine-linked (*O*-linked) glycosylation. By applying our optimized methods to increase concentrations of vesicle-bound glycosylation machinery, we shorten the time associated with extract preparation, increase glycosylation efficiencies, and enhance glycoprotein titers by up to ~170%. Importantly, we go on to show that improvements in glycoprotein titers are generalizable to multiple glycoproteins without the need to re-optimize conditions.

## Results

**Overview of results**. In this study, we aimed to characterize and engineer membrane vesicles (which form upon fragmentation of cell membranes during cell lysis) in *E. coli* CFE extracts (Fig. 1). Then, we used this knowledge to control enrichment of membrane-bound components for enhancing defined function, including improving glycoprotein synthesis (Fig. 1). To achieve these goals, we: (i) used nanocharacterization techniques to determine the sizes and quantities of membrane vesicles in *E. coli* extracts; (ii) determined how extract processing can control the enrichment of vesicles in extracts; (iii) enriched several heterologous, membrane-bound components in extracts via vesicles; and (iv) demonstrated that increasing enrichment of membrane-bound components significantly improves CFGpS systems for *N*- and *O*-linked glycosylation. This work sets the stage for portable biomanufacturing platforms that can broaden access to medicines by making them when and where they are needed.

**Characterization of membrane vesicles in CFE extracts**. Initially, we used several nanocharacterization techniques to analyze the size of vesicles and to visualize these particles in CFE extracts prepared using homogenization and 30,000g clarification (i.e., S30

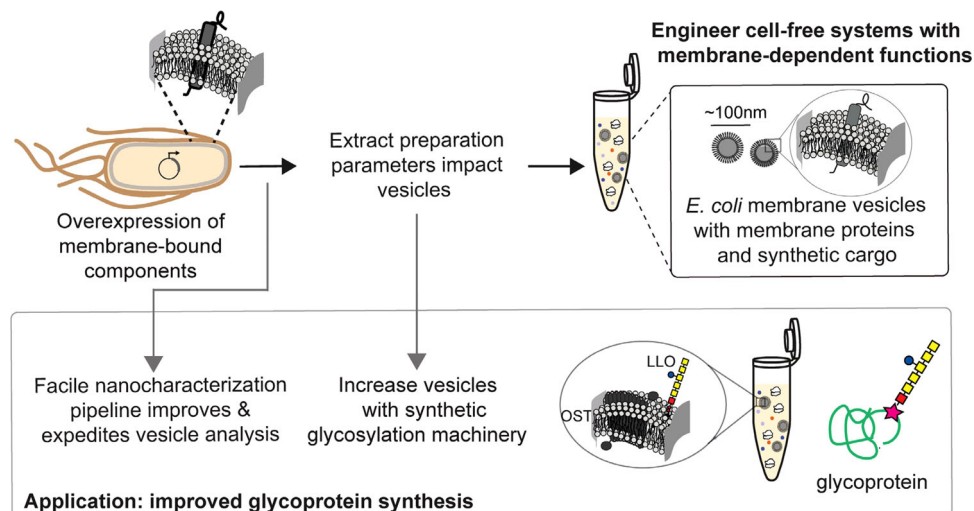

**Fig. 1 A platform for engineering cell-free gene expression (CFE) systems with cell-derived membrane-dependent functions.** Membrane-bound cargo expressed in living *E. coli* is carried through into CFE extracts via membrane vesicles. The extract preparation method used to prepare CFE extracts impacts sizes and concentrations of vesicles, and their associated cargo. Here, we develop a facile nanocharacterization pipeline to better understand and characterize the impacts of extract preparation methods on vesicle profiles and their associated cargo. We then apply our findings to improve cell-free glycoprotein synthesis (CFGpS), which is a promising platform for on-demand vaccine development. By increasing concentrations of vesicles and membrane-bound glycosylation machinery, oligosaccharyltransferases (OSTs) and lipid-linked oligosaccharides (LLOs), we overcome limitations in CFGpS and increase glycoprotein titers.

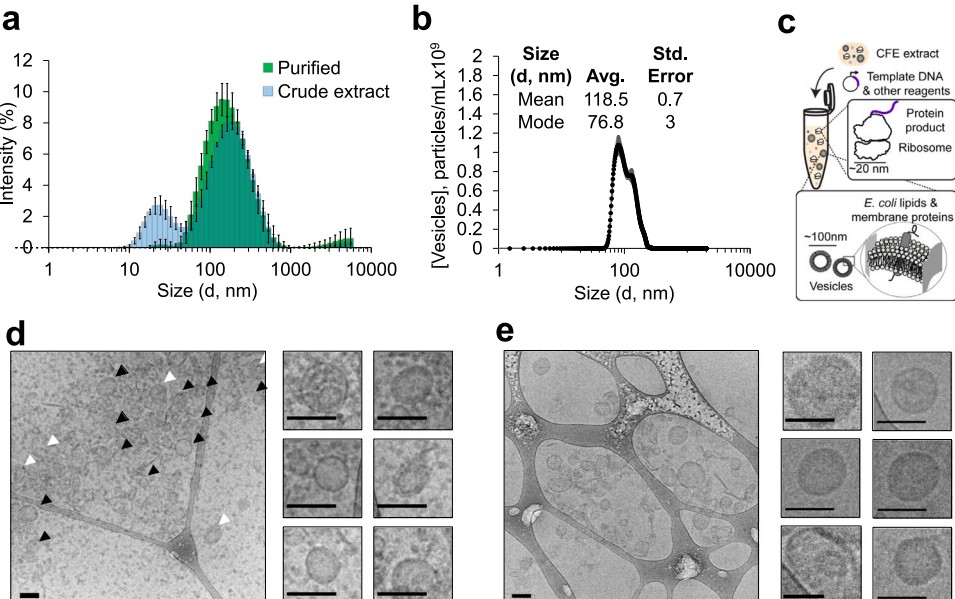

**Fig. 2 Characterization of membrane vesicles in crude CFE extracts. a** DLS analysis of crude extracts (blue) and SEC purified vesicles (green). Crude extract data are presented as mean intensity values for a given size ± standard deviation (SD) of $n = 9$ measurements (3 biologically independent extracts, each measured 3 times). Purified vesicle data are presented as mean intensity values for a given size ± SD of $n = 3$ measurements (one purified fraction measured 3 times). **b** NTA of purified vesicles collected from SEC. Mean and mode diameters observed in the particle size distribution are listed in the inset. Data are presented as mean ± standard error of the mean (SEM) of $n = 5$, 1 min NTA measurements of purified vesicles. **c** Illustration of particles detected in crude CFE extracts. **d** Cryo-EM micrographs of crude extracts. Black arrows indicate vesicles with apparent unilamellar morphology. White arrows indicate nested or multilamellar morphologies. Cropped images indicate representative vesicles. Scale bars are 100 nm. Uncropped images are available in Supplementary Fig. 3 and numbered with the corresponding cropped vesicles. **e** Cryo-EM micrographs of SEC purified vesicles. Cropped images indicate representative purified vesicle particles. Scale bars are 100 nm. Uncropped images are available in Supplementary Fig. 3 and numbered with the corresponding cropped vesicles. Micrographs are representative of three independent experiments. Source data for all panels are provided as a Source Data file.

extracts)[34]. Dynamic light scattering (DLS) analysis of crude extract revealed two major peaks: one narrower peak with an intensity maximum at ~20 nm, and a broader peak at ~100–200 nm (Fig. 2a). The 20 nm peak likely represents small cell-derived particles. *E. coli* ribosomes, which are present at ~1 µM in typical CFE reactions and enable the production of protein in our CFE reactions (Supplementary Fig. 1), are ~20 nm in size and likely contribute considerably to the signal measured[73,74]. We hypothesized that particles measured in the ~100–200 nm peak were vesicles. To directly analyze membrane vesicles without ribosomes and other cellular particles, we identified and purified membranous particles via size-exclusion chromatography (SEC)[75–77] (Supplementary Fig. 2A). DLS analysis of purified membrane vesicles revealed an intensity particle size distribution that directly overlapped with the proposed vesicle peak from our DLS traces of crude extracts (Fig. 2a). Nanoparticle tracking analysis (NTA), an orthogonal method for sizing and quantitating nanoparticles in solution, showed an average purified vesicle diameter of $118.5 \pm 0.7$ nm, corroborating the approximate size range of vesicles measured with DLS (Fig. 2b). The zeta potential of purified vesicles was $-14.5 \pm -1.0$ mV, indicating a negative particle surface charge consistent with phospholipid vesicles (Supplementary Fig. 2B). An illustration of particles detected in extract is shown in Fig. 2c.

Cryo-electron microscopy (cryo-EM) of extracts showed small ($\leq 20$ nm) particles and other larger, circular particles consistent with vesicle morphology (Fig. 2d). Cryo-EM micrographs of extracts revealed vesicles between ~40 nm and ~150 nm in size, and we observed intact vesicle morphologies both pre- and post-SEC purification (Fig. 2d, e). Uncropped and annotated cryo-EM micrographs are provided (Supplementary Fig. 3). Comparisons between measurements reveal that DLS, a bulk, in-solution measurement, overestimates vesicle diameter. DLS, however, is a useful tool for quickly characterizing crude extract particle profiles because it can detect particles <50 nm (including ribosomes) that are smaller than vesicles and are below the size limit of detection of NTA. Together, these results show particle profiles of crude extracts and indicate that vesicles are polydisperse, are on the order of tens to hundreds of nm across, and are relatively low in concentration compared with ribosomes and other small complexes.

**Extract processing impacts vesicle size distributions and concentrations**. To understand how to control membrane vesicles in extracts, we next sought to study how protocols to process extracts impacted vesicle properties. Specifically, we studied cell lysis and extract centrifugation because cell membranes are ruptured during lysis, and centrifugation dictates particle separation. We lysed cells using standard sonication (constant input energy per volume of cell suspension) or homogenization protocols (~20,000 psig)[11,34], then subjected lysates to a traditional 30,000*g* centrifugation protocol (termed 'S30' extracts), or a lower g-force protocol where the maximum centrifugation speed was 12,000*g* (termed 'S12' extracts) (Fig. 3a)[11,12]. These combinations of lysis and centrifugation protocols were selected because they have previously been used to obtain high-yielding *E. coli* CFE extracts[78]. Indeed, all the conditions tested yielded extracts that were active for protein synthesis in standard CFE reaction conditions (Supplementary Fig. 4A). The combination of a standard homogenization and S30 prep represents our base case because extracts used in our previously described one-pot CFGpS platform were prepared with these conditions, as well as the extracts used in our initial characterization here (Fig. 2). Before this work, S12 extracts had not previously been used for making glyco-competent CFE extracts.

Of the conditions tested, the centrifugation protocol had the most impact on vesicle concentrations. We observed higher numbers of vesicles in S12 extracts for both lysis methods, with the reduced centrifugation speed likely being the reason for increased particle concentrations. Specifically, we observed 1.2- and 2.0-fold enrichments of vesicles in sonicated and homogenized S12 extracts, respectively (Fig. 3b). Homogenized S12 extracts contained the highest concentration of vesicles with $6.5 \pm 0.3 \times 10^{12}$ particles/mL (as compared to $3.4 \pm 0.1 \times 10^{12}$ particles in the base case), making it the most promising condition for enriching vesicles.

While centrifugation impacted vesicle concentration, lysis method impacted vesicle size. Sonicated extracts contained smaller vesicles with narrower size distributions than homogenized extracts, regardless of centrifugation protocol. Our observations that lysis method impacts vesicle size is consistent with studies showing that varying experimental parameters to disperse phospholipids (or amphiphiles in general) impacts vesicle sizes[79]. Particle size distributions of sonicated extracts reached single maxima at ~110 nm, with average particle diameters of ~130 nm; homogenized extracts had higher average particle diameters of ~160 nm, displaying distinct peaks at ~120 nm, and considerable shoulder peaks at ~150 nm (Fig. 3c, d and Supplementary Fig. 5A). The particle size distributions observed in homogenized extracts may indicate the presence of multiple, discrete, vesicle populations (Fig. 3c, d). DLS measurements confirmed the observation that sonicated extracts contained relatively smaller, less polydisperse vesicles than homogenized extracts (Supplementary Fig. 5B, C). Notably, direct vesicle analysis in extracts enabled us to gauge the impacts of extract processing in ways that have not been previously accessible and provides benchmarks for intact vesicle concentrations in extracts.

**Heterologous membrane-bound cargo can be controllably enriched via membrane vesicles**. With a better understanding of the characteristics and concentrations of native vesicles, we sought to enrich extracts with vesicles containing heterologous cargo derived from the periplasmic membrane of *E. coli*. Since S12 extracts contain higher concentrations of vesicles than S30 extracts, we hypothesized that S12 extracts would also contain higher concentrations of associated heterologous cargo. The highest dynamic range of vesicle concentration between S12 and S30 preparations was observed with homogenization, so we proceeded with homogenization for enrichment experiments (Fig. 3b). We overexpressed six membrane-bound proteins of various sizes, transmembrane topologies, biological functions, and taxonomical origins to test for enrichment (Supplementary Table 1). The proteins selected for enrichment encompass classes of proteins that could enable expanded functionalities in CFE, including glycosylation enzymes (PglB, PglO, STT3) and signal transduction/sensing proteins (NarX, PR, CB1). We expressed each membrane protein in vivo with a C-terminal FLAG tag, prepared S30 and S12 extracts, then analyzed concentrations of the overexpressed membrane protein using quantitative western blotting. We observed approximately 2-fold membrane protein enrichment in S12 over S30 (S12/S30) extracts for all proteins other than PR, for which we observed ~4-fold enrichment (Fig. 4a, b). As a control, when sfGFP with no transmembrane helices was expressed in vivo, we did not observe significant S12/S30 enrichment (Fig. 4c). Full blots for Fig. 4a–c are shown in Supplementary Fig. 6. Notably, enrichment values obtained via blotting correspond closely with the 2-fold vesicle enrichment observed via NTA in homogenized S12 and S30 extracts with no overexpression (Fig. 3b). All extracts with pre-enriched

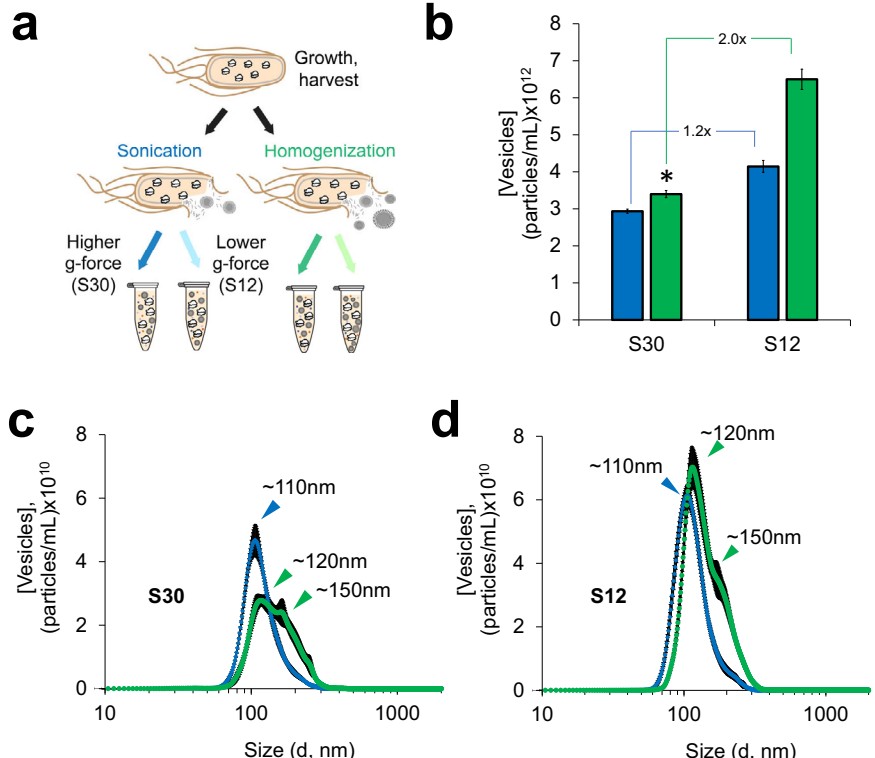

**Fig. 3 Extract processing impacts vesicle size distributions and concentrations. a** Illustration of extract processing conditions. Extracts were prepared in triplicate for each condition shown. **b** Nanoparticle Tracking Analysis (NTA) concentration analysis of vesicles in sonicated (blue) and homogenized (green) extracts. Asterisk indicates base case conditions for extract preparation. Data for (**b**) are presented as mean ± SD of $n = 15$ replicates (3 biologically independent extracts; each examined over 5, 1 min NTA measurements). **c** NTA particle size distribution of vesicles in sonicated (blue) and homogenized (green) S30 extracts. **d** NTA particle size distribution of sonicated (blue) and homogenized (green) S12 extracts. Data for (**c, d**) are presented as mean ± SEM of $n = 15$ replicates (3 biologically independent extracts; each examined over 5, 1 min NTA measurements). Source data for all panels are provided as a Source Data file.

membrane proteins displayed protein synthesis activity (Supplementary Fig. 4B).

With an eye towards bacterial glycoengineering applications, we next confirmed that PglB and PglO, key enzymes for glycosylation, were associated with membrane vesicles, as opposed to free in solution (Fig. 4d). Extracts with pre-enriched PglB or PglO were probed with a green fluorescent α-FLAG antibody, then analyzed via SEC. Fluorescence chromatograms are shown in Fig. 4d, with the characteristic vesicle elution fraction highlighted in gray (Supplementary Fig. 2A). The characteristic vesicle elution peak corresponded with green fluorescence for extracts containing PglB or PglO and no corresponding peak was observed in an extract with no overexpressed membrane protein (Fig. 4d). Our results show that heterologous cargo that is embedded in the periplasmic (inner) membrane of *E. coli* cells can be pre-enriched in extract and tuned via vesicles.

**Increasing vesicle concentrations improves cell-free glycoprotein synthesis (CFGpS) for *N*- and *O*-linked glycosylation systems.** We next set out to exploit our ability to enrich vesicles harboring heterologous cargo in an application. We focused on protein glycosylation, because glycosylation plays critical roles in cellular function, human health, and biotechnology. As a model, we sought to increase glycoprotein yields in a previously reported CFGpS platform by charging reactions with S12 extracts containing higher concentrations of membrane-bound glycosylation machinery[34]. We prepared S30 and S12 extracts from strains overexpressing the model *N*-linked glycosylation pathway from

*Campylobacter jejuni*, which consists of the membrane-bound oligosaccharyltransferase (OST) PglB that catalyzes glycosylation, and a lipid-linked oligosaccharide (LLO) donor of the form: GalNAc-α1,4-GalNAc-α1,4-(Glcβ1,3)-GalNAc-α1,4-GalNAc-α1,4-GalNAc-α1,3-Bac (where Bac is 2,4-diacetamido-2,4,6-tri-deoxyglucopyranose) from an undecaprenyl-pyrophosphate-linked donor[80]. NTA and western blot analysis of CFGpS extracts revealed 2.5-fold S12/S30 enrichment of vesicles and a corresponding 2-fold S12/S30 enrichment of PglB (Supplementary Fig. 7). Fluorescence staining and SEC analysis confirmed the presence and association of LLO and PglB with the vesicles (Supplementary Fig. 8A).

To assess the impact of enriched vesicles on CFGpS, we carried out reactions in two phases (Fig. 5a, inset)[23]. First, cell-free protein synthesis (CFPS) of the acceptor protein was run for a defined time, termed 'CFPS time'. At the CFPS time, reactions were spiked with MnCl$_2$, quenching CFPS and initiating glycosylation by providing the OST with its Mn$^{2+}$ cofactor. CFGpS reactions charged with S30 or S12 extracts were run for CFPS times of 2, 10, 20, 30, and 60 min using a His-tagged sfGFP$_{DQNAT}$ acceptor protein, where DQNAT is a permissible PglB sequon (with N being the glycosylated residue). Coding sequences of all acceptor proteins used are presented in Supplementary Table 2. Endpoint glycoprotein yields were quantified using total acceptor protein yield, determined by sfGFP fluorescence and $^{14}$C incorporation, and % glycosylation, determined by western blotting (Fig. 5a and Supplementary Fig. 9A–D). At longer CFPS times, we observed that S12 extracts produced significantly more glycoprotein than S30 extracts. Because total acceptor protein concentrations for S30 and S12

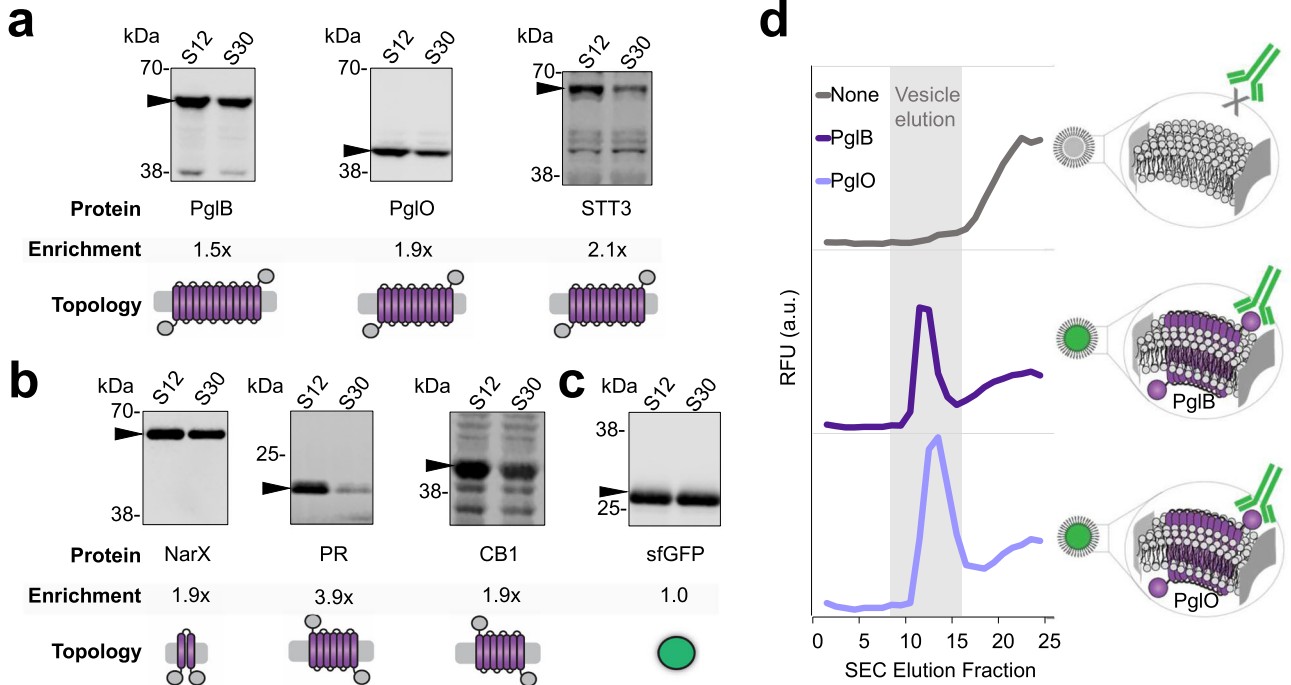

**Fig. 4 Heterologous membrane-bound cargo can be controllably enriched via membrane vesicles.** Enrichment of heterologous membrane proteins in S12 and S30 extracts was quantitated using α-FLAG western blots against the heterologous proteins for: **a** glycosylation enzymes, **b** signal transduction proteins, and **c** a cytosolic sfGFP control with no transmembrane helices. On each western blot, left lanes are S12 extracts and right lanes are S30 extracts. Black arrows indicate the membrane protein of interest. Molecular weight (kDa) from protein ladder standards are indicated to the left of each blot. Protein names and enrichment ratios of bands (S12/S30) are shown directly below each blot. All blots are representative of $n = 3$ biologically independent extracts. Cartoons depict the transmembrane topology for each protein. See Supplementary Table 1 for taxonomical origin, transmembrane topology, functions(s), theoretical size, and UniProt ID. Full western blot images for panels (**a**–**c**) are available in Supplementary Fig. 6 and Source Data file. **d** Fluorescence chromatograms of SEC analysis of extracts probed with a fluorescent α-FLAG antibody. Strains used to prepare extracts were enriched with no membrane protein (gray trace), PglB (dark purple trace), or PglO (light purple trace). Characteristic vesicle elution fraction from 3 independent experiments is highlighted in gray. Source data for all panels are provided as a Source Data file.

reactions were similar for each CFPS time (Supplementary Fig. 9E), increased glycoprotein yield in S12 extracts is due to higher glycosylation activity and not higher CFE yields. Specifically, at 20, 30, and 60 min CFPS times, we observed 67%, 85%, and 91% increases in glycoprotein yield in the S12 reactions, respectively. At the 60 min CFPS time, S12 reactions yielded $117.2 \pm 9.9$ µg/mL of glycoprotein in batch (Fig. 5a). Notably, these optimizations enabled batch glycoprotein titers on the order of hundreds of µg/mL in a crude-extract-based CFGpS system without extra vesicle supplementation to the reactions[2]. This advance was enabled by using S12 extracts instead of S30 extracts, and relying on enriched cell-derived membranes. S12 reactions also had significantly higher terminal % glycosylation, or percent of CFPS-derived acceptor protein that is glycosylated at the end of a 16 h glycosylation reaction. This was true for all CFPS times that were tested (Supplementary Fig. 9F). For example, for reactions with 20 min CFPS times, we observed an increase from 51% glycosylation for S30 reactions to 82% glycosylation for S12 reactions (Fig. 5b and Supplementary Fig. 9F). Western blots of representative reactions using α-His (showing glycosylated and aglycosylated acceptor protein) and α-glycan (against the *C. jejuni* glycan) are shown in Fig. 5c. Taken together, these results indicate that the higher concentration of membrane-associated glycosylation components in S12 extracts has a measurable effect on CFGpS, improving glycoprotein yields and endpoint % glycosylation.

With a long-term interest in synthesizing diverse glycoproteins in cell-free systems, we next ported an *O*-linked glycosylation system, known to have broad glycan specificity, into the CFGpS

platform[72,81,82]. We selected the *O*-OST PglO from *Neisseria gonorrhoeae* that accepts the *C. jejuni* heptasaccharide LLO as a donor, but differs from PglB in acceptor sequence preferences[83]. For PglO, we used an sfGFP-fusion acceptor protein containing a recently determined 8 amino acid (WPAAASAP, with S being the glycosylated residue) minimum optimal *O*-linked recognition site (termed 'MOOR')[83]. We confirmed residue-specific *O*-linked glycosylation and enrichment of PglO and LLO in vesicles (Supplementary Figs. 10 and 8B). As additional proof of site-specific glycosylation, we performed liquid chromatography mass spectrometry (LC-MS/MS) analysis of the glycoproteins obtained via CFGpS with PglO and PglB and observed the presence of the 1406 Da *C. jejuni* heptasaccharide on the expected tryptic peptides (Supplementary Fig. 11A, B)[80]. As in PglB-mediated CFGpS, we observed increased endpoint glycoprotein yield and % glycosylation in reactions charged with S12 extracts. Specifically, reactions with CFPS times of 20 min resulted in a 69% increase in glycoprotein yield and an increase from 27% to 40% glycosylation in reactions with S12 extracts compared to those containing S30 extracts (Fig. 5d and Supplementary Fig. 12). Corresponding blots are shown in Fig. 5e and Supplementary Fig. 12A, B. Collectively, these results indicate that improvements to glycosylation in S12 extracts translate from the *N*-linked glycosylation system to the *O*-linked glycosylation system.

To determine whether enhanced glycoprotein production in S12 extract-based CFGpS reactions was transferrable to non-model acceptor proteins, we tested three additional proteins. This included the *C. jejuni* AcrA, a native bacterial glycoprotein with two internal glycosylation sites[50,84], as well as two possible carrier

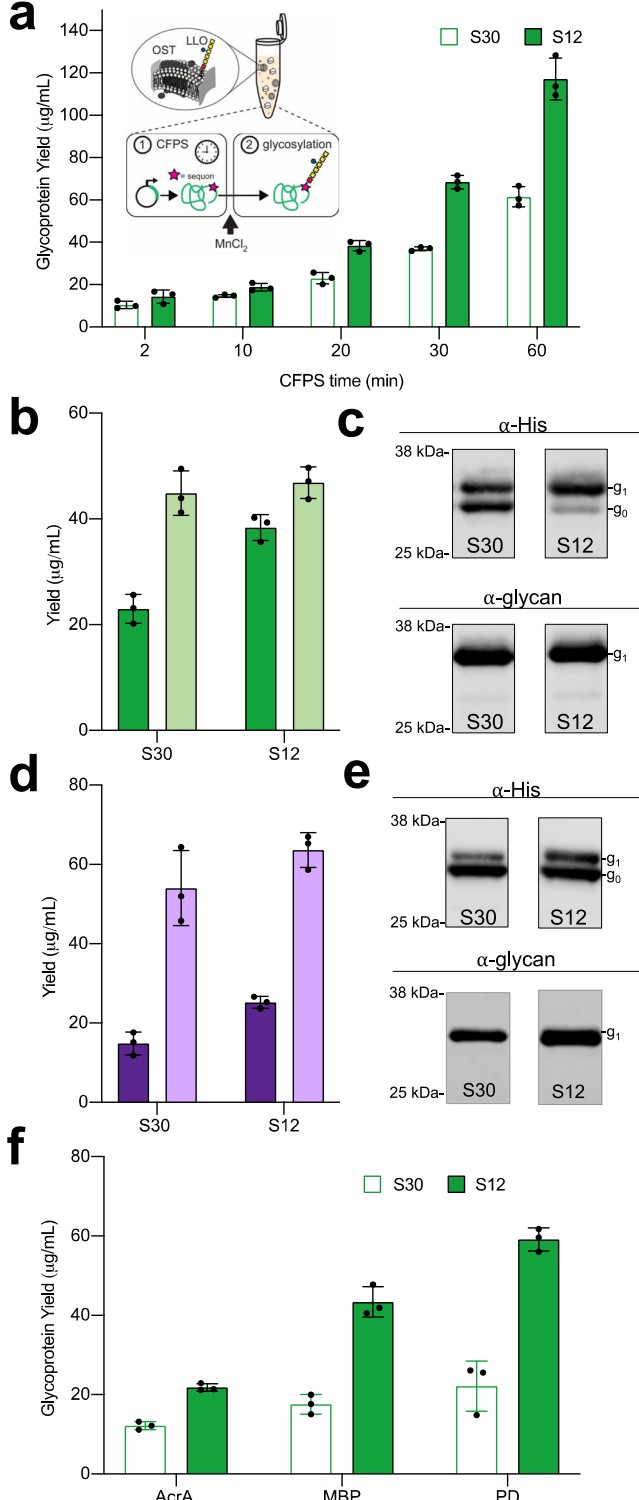

**Fig. 5 Increasing vesicle concentrations improves cell-free glycoprotein synthesis (CFGpS) for N- and O-linked glycosylation systems.** For panels (**a**–**e**), a standard curve correlating protein yields derived from [14]C-leucine counting and sfGFP fluorescence was used to measure total protein concentrations. Quantitative western blotting was used to measure fraction of glycosylated protein. For panel (**f**), protein concentrations were measured using [14]C-leucine incorporation. Fraction of glycosylated protein was measured using autoradiography. **a** sfGFP glycoprotein yields of CFGpS reactions charged with S12 (green) or S30 (white) extracts enriched with PglB and *C. jejuni* LLO. Data are presented as mean ± SD of n = 3 biologically independent CFGpS reactions. (Inset) Schematic of two-phase CFGpS reactions. **b** Glycosylated (dark green) and total (light green) sfGFP yields of *N*-linked CFGpS reactions with 20 min CFPS times. Data are presented as mean ± SD of n = 3 biologically independent CFGpS reactions. **c** Anti-His and anti-glycan western blots of acceptor proteins from representative reactions in (**b**) show glycosylated (g1) and aglycosylated (g0) protein. Full western blot images are available in Supplementary Fig. 9A–D and Source Data file. **d** Glycosylated (dark purple) and total (light purple) sfGFP yields from *O*-linked CFGpS reactions with 20 min CFPS times. Data are presented as mean ± SD of n = 3 biologically independent CFGpS reactions. **e** Anti-His and anti-glycan western blots of acceptor proteins from representative reactions in (**d**) show glycosylated (g1) and aglycosylated (g0) protein. Full western blot images are available in Supplementary Fig. 12A, B and Source Data file. **f** Glycoprotein yields of AcrA, MBP, and PD produced in CFGpS reactions charged with S12 (green) or S30 (white) extracts enriched with PglB and *C. jejuni* LLO. Glycoprotein yields of AcrA, which contains 2 internal glycosylation sites, include singly and doubly glycosylated protein. Data are presented as mean ± SD of n = 3 biologically independent CFGpS reactions. Source data for all panels are provided as a Source Data file.

MBP, and PD, respectively, when comparing S12- to S30-based reactions. Expression improvements were determined by [14]C-leucine incorporation (Fig. 5f and Supplementary Fig. 13A–E). Our results highlight that the improvements in glycoprotein yield observed in extracts with higher concentrations of vesicles hold for diverse proteins without the need for re-optimization.

## Discussion

In this work, we set out to benchmark, understand, and quality-control protein-enriched vesicles in bacterial CFE extracts for expanding and enhancing functionality. We showed that upstream extract processing can be used to tune concentrations of vesicles and associated cargo from the periplasm. Then, we applied this knowledge to improve CFGpS, with a specific application focus of glycoprotein synthesis. Our results have several key features.

First, the light scattering tools used here allowed us to quickly quantify intact vesicle numbers and sizes in CFE extracts. This is important because this knowledge informed design rules for enhancing vesicle concentrations and functionality from their associated protein cargo in cell-free systems. Notably, the effective vesicle surface area calculated from NTA measurements (~0.3 m$^2$ membrane/mL extract) is consistent with values calculated from phospholipid concentrations in similar extracts[56].

Second, our results offer insights into field-wide observations and limitations of *E. coli*-based CFE systems. For example, it is well-documented that lysis protocols can impact CFE productivity[87]. Our findings show that lysis methods impact size distributions of vesicles generated during this step, which affect the membrane environment of the machinery necessary for oxidative phosphorylation and ATP regeneration. Since vesicles are key for activating cost-effective energy metabolism from oxidative phosphorylation in CFE, routine vesicle characterization could

proteins for conjugate vaccines. The possible carrier proteins were: *Haemophilus influenzae* protein D (PD), which is a licensed carrier protein, and *E. coli* maltose-binding protein (MBP), which is not yet a licensed carrier but has shown promising results in clinical studies[85,86]. PD and MBP were each fused to a single C-terminal DQNAT sequon to enable glycosylation[23]. CFGpS was run as previously described using a 20 min CFPS time. All conditions were held constant other than the DNA template and addition of [14]C-leucine to enable quantification. We observed 80%, 147%, and 167% increases in glycoprotein titers for AcrA,

become a vital quality-control check, leading to improved reproducibility in and between labs[88]. Our results also offer insight into why, despite the presence of vesicles in the *E. coli* CFE system, CFE-derived membrane proteins cannot be synthesized via insertion into native vesicles without additional vesicle supplementation[48,50,54]. With ~6 nM of intact vesicles in CFE reactions (where intact vesicle concentration was calculated from NTA measurements), the concentration of vesicles is orders of magnitude lower than typical protein titers produced in our CFE extracts (~30 µM of reporter protein or higher).

Third, our work opens the door to engineering cell-free systems that rely on enriched membrane-bound components. We show that membrane-bound proteins and LLOs expressed in vivo in the periplasm can be enriched in vesicles, indicating that a population of vesicles is derived from the inner periplasmic membrane[89,90]. Importantly, our workflow easily interfaces with numerous methods that could be used to alter vesicles and their membrane-bound cargo. For example, using other centrifugation speeds besides 12,000*g* could result in changes to vesicle concentration. In addition, additives could be supplemented to cell-free systems to tune biophysical features of membrane properties (e.g., composition, size, fluidity, curvature). Furthermore, unlike the previously used S30 extract procedure, the optimized S12 extract strategy developed here does not require a high-speed centrifuge and is less time-intensive. This simplifies the CFGpS platform, enabling the process from inoculation of cell culture to testing CFGpS reactions to be completed in a single workday. And, while we focus entirely on *E. coli*-based systems here, the reported characterization methods could, in principle, be extended to further optimize insect and CHO-based CFE systems that rely on ER-derived microsomes to perform glycosylation, embed nascent membrane proteins, and perform other membrane-dependent functions.

Towards applications in biomanufacturing, a key feature of the *E. coli*-based CFGpS system is expressing synthetic glycosylation pathways encoding diverse O-antigens from pathogenic bacteria. This feature points toward immediate utility of our CFGpS system in the on-demand bioproduction of conjugate vaccines[23]. Here, we show that S12 extracts enable higher glycoprotein titers of two glycoconjugate vaccine carrier proteins modified with a model *C. jejuni* LLO, indicating that vaccine production may be simpler and more efficient using the optimized methods reported here. Additionally, we have recently shown that our optimized S12 conditions can be used to recapitulate efficient, humanized *O*-linked glycosylation in glycoengineered *E. coli* extracts[72]. While applications in *O*-linked glycosylation and conjugate vaccines are imminent, the recapitulation of efficient eukaryotic-type *N*-linked glycosylation (i.e., glycoproteins with a $Man_3GlcNAc_2$ core glycan) for production of therapeutics still remains on the horizon in *E. coli*-based systems.

Future studies to elucidate translocation and co-translational glycosylation in vesicles will be important. These studies could be especially useful for producing complex, native glycoproteins for which protein glycosylation and folding are co-translational. While it has been shown that glycosylation with PglB can proceed on pre-folded proteins in vitro (using purified, reconstituted components and without the need for translocation or intact membranes[91]), obtaining a more robust understanding of the topology of glycosylation in membrane vesicles is an important future effort for therapeutics production.

Looking forward, we anticipate that our work will accelerate efforts to manufacture proteins that require membrane-dependent modifications, such as glycoproteins. For example, the approach described enables *N*-linked glycoprotein synthesis yields of >100 µg/mL, which increases accessibility for on-demand vaccine production in resource-limited settings. In sum, our results pave the way for efficient, accessible CFE systems that require membrane-bound activities for expanding system functionality and enabling a variety of synthetic biology applications.

## Methods

**Extract preparation**. The chassis strain used for all extracts was CLM24[34]. Source strains were grown in 1 L of 2×YTPG media at 37 °C with agitation. Cells were grown to OD 3, then harvested by centrifugation (5000*g*, 4 °C, 15 min). For overexpression of proteins in vivo, CLM24 source strains were grown at 37 °C in 2xYTPG with the appropriate antibiotic(s), listed in Supplementary Table 3. Cells were induced with 0.02% (wt/vol.%) L-arabinose at OD 0.6–0.8, shifted to 30 °C, and harvested at OD 3. All subsequent steps were carried out at 4 °C and on ice unless otherwise stated. Pelleted cells were washed 3 times in S30 buffer (10 mM Tris acetate pH 8.2, 14 mM magnesium acetate, 60 mM potassium acetate). After the last wash, cells were pelleted at 7000*g* for 10 min, flash-frozen and stored at −80 °C. After growth and harvest, cells were thawed and resuspended to homogeneity in 1 mL of S30 buffer per gram of wet cell mass. For homogenization, cells were disrupted using an Avestin EmulsiFlex-B15 high-pressure homogenizer at 20,000–25,000 psig with a single pass (Avestin, Inc. Ottawa, ON, Canada). For sonication, input energy was calculated using an empirical correlation[11]. Cells were sonicated on ice using a Q125 Sonicator (Qsonica, Newtown, CT) with a 3.175 mm diameter probe at a frequency of 20 kHz and 50% of amplitude. Energy was delivered to cells in pulses of 45 s followed by 59 s off until the target energy was delivered. Cells were lysed and clarified in triplicate. For S30 preparation, lysed cells were centrifuged twice at 30,000*g* for 30 min; supernatants were transferred to a fresh tube for each spin. Supernatants were incubated with 250 rpm shaking at 37 °C for 60 min for runoff reactions. Following runoff, lysates were centrifuged at 15,000*g* for 15 min. Supernatants were collected, aliquoted, flash-frozen, and stored at −80 °C for further use. For S12 preparation, lysed cells were centrifuged once at 12,000*g* for 10 min; supernatants were collected and subjected to runoff reactions as described above. Following runoff, lysates were centrifuged at 10,000*g* for 10 min at 4 °C. Supernatants were collected, aliquoted, flash-frozen in liquid nitrogen, and stored at −80 °C.

**Dynamic light scattering (DLS) and nanoparticle tracking analysis (NTA) measurements**. DLS measurements were performed on a Zetasizer Nano ZS (Malvern Instruments Ltd., UK) with a measurement angle of 173° in disposable cuvettes (Malvern Instruments Ltd., UK ZEN0040). All measurements were collected in triplicate for 13 scans per measurement. Refractive index and viscosity were obtained from the instrument's parameter library. The instrument's 'General Purpose' setting was used to calculate intensity and number particle size distributions. For DLS of crude extracts, extracts were diluted 1:10 with 0.1 µm filtered PBS before analysis. For purified vesicle samples, elutions were analyzed directly without dilution.

NTA measurements were performed on a Nanosight NS300 using a 642 nm red laser (Malvern Instruments Ltd., UK). Samples were diluted to manufacturer-recommended particle concentrations in sterile PBS until a linear trend between dilution factor and concentration measured was found. Samples were flowed into the cell, and the instrument was focused according to manufacturer recommendations. Measurements were collected at room temperature, using a 1 mL syringe and a syringe pump infusion rate of 30 (arbitrary units). Data for each sample was collected in 5 separate 1 min videos, under continuous flow conditions. Mean particle diameters and particle concentrations were obtained from aggregate Nanosight experiment reports of each run, then averaged across triplicates and corrected for dilution factor.

**Transmission electron microscopy (TEM)**. For cryo-TEM measurement, 200 mesh Cu grids with a lacey carbon membrane (EMS Cat. # LC200-CU) were placed in a Pelco easiGlow glow discharger (Ted Pella Inc., Redding, CA, USA) and an atmosphere plasma was introduced on the surface of the grids for 30 s with a current of 15 mA at a pressure of 0.24 mbar. This treatment creates a negative charge on the carbon membrane, allowing for aqueous liquid samples to spread evenly over the grid. Then, 4 µL of sample was pipetted onto the grid and blotted for 5 s with a blot offset of +0.5 mm, followed by immediate plunging into liquid ethane within a FEI Vitrobot Mark III plunge freezing instrument (Thermo Fisher Scientific, Waltham, MA, USA). Grids were then transferred to liquid nitrogen for storage. The plunge-frozen grids were kept vitreous at −172 °C in a Gatan Cryo Transfer Holder model 626.6 (Gatan Inc., Pleasanton, CA, USA) while viewing in a JEOL JEM1230 LaB6 emission TEM (JEOL USA, Inc., Peabody, MA) at 120 keV. Image data were collected by a Gatan Orius SC1000 CCD camera Model 831 (Gatan Inc., Pleasanton, CA, USA). Image analysis was done using ImageJ.

**Plasmid construction**. All plasmids used in this study are listed in Supplementary Table 3. Supplementary Table 4 includes primers used to clone plasmids constructed for this study. pSF-NgPglO was amplified from an existing plasmid using primers 1 and 2 (Supplementary Table 4). gBlocks for PR, HsCB1, LmSTT3D, and sfGFP were ordered with homology to the pSF backbone. pSF plasmids were then

assembled using Gibson Assembly to combine gBlocks and pSF backbone, which was amplified using primers 3 and 4 (Supplementary Table 4). sfGFP-MOOR and sfGFP-MOORmut were constructed using Gibson Assembly to assemble sfGFP and MOOR/MOORmut-pJL1. sfGFP with homology to pJL1 and the linker (used between MOOR/MOORmut and sfGFP) was amplified using primers 5 and 6 (Supplementary Table 4). PJL1 with the MOOR or MOORmut tag and linker (used between tag and sfGFP) was amplified using primers 7 and 8 (Supplementary Table 4).

**Western blotting and densitometry analyses.** SDS-PAGE was run using NuPAGE 4–12% Bis-Tris protein gels with MOPS-SDS buffer (Thermo Fisher Scientific, Waltham, MA, USA). After electrophoresis, proteins were transferred from gels to Immobilon-P polyvinylidene difluoride 0.45 μm membranes (Millipore, USA) according to manufacturer's protocol. Membranes were blocked in either Odyssey or Intercept blocking buffer (LI-COR, USA) and washed with 1x PBST. α-FLAG blots of membrane proteins were probed using α-FLAG antibody (Abcam 2493) at a 1:5000 dilution in blocking buffer with 0.2% Tween20 (Sigma-Aldrich) as the primary. α-His blots were probed with 6xHis-antibody (Abcam, ab1187) at a 1:7500 dilution in blocking buffer with 0.2% Tween20 (Sigma-Aldrich) as the primary. For α-glycan blots, hR6 serum from rabbit that binds to the native *C. jejuni* glycan at a 1:5000 dilution in blocking buffer with 0.2% Tween20 (Sigma-Aldrich) was used as the primary probe[49]. A fluorescent goat α-Rabbit IgG IRDye 680RD (LI-COR, USA) at a 1:10,000 dilution in blocking buffer with 0.2% Tween20 (Sigma-Aldrich) and 0.01% SDS (Ambion) was used as the secondary for all blots. Blots were imaged using a LI-COR Odyssey Fc (LI-COR Biosciences, USA). Densitometry was performed using Image Studio Lite software to measure band intensity. Fluorescence background was subtracted from blots before determining band intensities. For determining membrane protein enrichment (S12/S30), band intensities of membrane proteins for three independent S12 extract replicates and three independent S30 replicates were measured for each protein. The rounded averages of triplicate ratios (S12/S30) and associated error are reported as enrichment in Fig. 4. For determining glycoprotein yields from CFGpS reactions, band intensities for glycosylated and aglycosylated bands were obtained from independent, triplicate reactions. The fraction of glycosylated protein for each replicate was calculated via band intensities. To obtain glycoprotein yields, the fraction glycosylated was multiplied by total protein yield for each replicate as calculated from sfGFP fluorescence converted to protein concentration (described below). Yields were plotted using Prism v.9.0.0 (GraphPad).

**Lipid dye staining and fluorescence immunostaining of vesicles.** All reagents used for immunostaining and SEC were sterile filtered with a 0.1 μm filter (Millex-VV Syringe Filter, Merck Millipore Ltd. or Rapid-Flow Filter, Nalgene). To determine vesicle elution fractions, extract was probed with FM 4-64 lipid dye (Life Technologies), a lipophilic styrene dye that has low fluorescence in aqueous solution and becomes brightly fluorescent upon incorporation into membranes. FM-464 dye preferentially stains the inner membrane of *E. coli*, but has been used to dye the outer membrane as well[92,93]. FM 4-64 lipid dye was prepared in stock solutions at 10 mg/mL in 100% DMSO, then diluted 1000-fold in nuclease-free water before use. Then, 80 μL of extract, 10 μL 10x PBS, and 10 μL of FM 4-64 were mixed to a final concentration of 1 ng dye/μL. Samples were incubated with dye in the dark for 10 min at 37 °C prior to SEC. To verify the presence of glycosylation components in vesicles, we probed for the LLO with a red fluorescent soybean agglutinin (SBA) lectin, a protein complex which specifically binds to the *C. jejuni* LLO[80], and for the OST with an orthogonal green fluorescent α-FLAG antibody as described above. For α-FLAG immunostaining and SBA staining, 90 μL extract and 10 μL of 10xPBS were mixed with 2 μL of α-FLAG-DyLight 488 (MA191878D488, Invitrogen, USA) and 4 μL of SBA-AlexaFluor™ 594 (32462, Invitrogen, USA). Antibody and SBA were incubated with extract in the dark with agitation overnight at 4 °C prior to SEC.

**Size-exclusion chromatography (SEC) of vesicles.** Here, 100 μL of extract mixture (stained with lipid dye or antibody) was flowed over a SEC column with PBS. Elution fractions were collected into a clear polystyrene 96-well plate (Costar 3370, Corning Inc., USA) at a rate of 0.4 min/well using a Gilson FC 204 Fraction Collector (Gilson, Inc., USA). Poly-Prep chromatography columns (Bio-Rad, USA) were packed with 8 mL of Sepharose 4B resin 45–165 μm bead diameter, (Sigma Aldrich, USA) and washed with sterile PBS 3 times before use. Elution fluorescence was measured using a Synergy H1 microplate reader (BioTek, USA). Excitation and emission wavelengths for SBA-AlexaFluor™ 594 were 590 and 617 nm, respectively. Excitation and emission wavelengths for α-FLAG-DyLight 488 were 493 and 528 nm, respectively. Vesicles stained with FM 4-64 lipid dye were used to determine the characteristic vesicle elution fraction. Reference samples probed with FM 4-64 were used to determine the characteristic vesicle elution fraction in each experiment. For plots, SBA curves were background subtracted.

**CFE reactions.** Protein synthesis was carried out with a modified PANOx-SP system in triplicate reactions, with each reaction containing a uniquely prepared extract[87]. Specifically, 1.5 mL microcentrifuge tubes (Axygen, MCT-150-C) were charged with 15 μL reactions containing 200 ng pJL1-sfGFP plasmid (Supplementary Table 1), 30% (vol./vol.%) extract, and the following: 6 mM magnesium glutamate (Sigma, 49605),

10 mM ammonium glutamate (MP, 02180595), 130 mM potassium glutamate (Sigma, G1501), 1.2 mM adenosine triphosphate (Sigma A2383), 0.85 mM guanosine triphosphate (Sigma, G8877), 0.85 mM uridine triphosphate (Sigma U6625), 0.85 mM cytidine triphosphate (Sigma, C1506), 0.034 mg/mL folinic acid, 0.171 mg/mL *E. coli* tRNA (Roche 10108294001), 2 mM each of 20 amino acids, 30 mM phosphoenolpyruvate (PEP, Roche 10108294001), 0.4 mM nicotinamide adenine dinucleotide (Sigma N8535-15VL), 0.27 mM coenzyme-A (Sigma C3144), 4 mM oxalic acid (Sigma, PO963), 1 mM putrescine (Sigma, P5780), 1.5 mM spermidine (Sigma, S2626), and 57 mM HEPES (Sigma, H3375). To gauge extract CFE productivity, reactions were carried out for 20 h at 30 °C.

**Quantification of CFE and CFGpS protein yields.** The concentration of cell-free-derived sfGFP was determined by measuring in-extract fluorescence and then converting to protein concentration using a standard curve relating sfGFP fluorescence to protein concentration as determined by a [14C]-leucine incorporation assay[34]. Briefly, 2 μL of cell-free reaction product was diluted into 48 μL of Ambion nanopure water (Invitrogen, USA). The solution was then placed in a Costar 96-well black assay plate (Corning Inc., USA). Fluorescence was measured using a Synergy H1 microplate reader (BioTek, USA) and Gen5 v. 2.09 (BioTek) software. Excitation and emission wavelengths for sfGFP fluorescence were 485 and 528 nm, respectively. This RFU value was then used to calculate the protein concentration. Yields were plotted using Prism v.9.0.0 (GraphPad).

Yields of all acceptor proteins (other than sfGFP) were assessed directly via the addition of 10 μM [14C]-leucine (PerkinElmer) to the CFGpS reaction to yield trichloroacetic acid-precipitable radioactivity that was measured via scintillation counting. Soluble fractions were isolated after centrifugation at ≥12,000g for 15 min at 4 °C. Briefly, 6 μL of the soluble fraction of CFGpS reactions run with 20 min CFPS times were mixed with 6 μL 0.5 M KOH and incubated for 20 min at 37 °C. 5 μL μL of treated sample was then soaked into two separate filtermats (PerkinElmer Printer Filtermat A 1450-421) and dried under a heat lamp. One filtermat was washed 3 times using 5% trichloroacetic acid (TCA) with 15 min incubations at 4 °C, and then once with ethanol with a 10 min incubation at room temperature. Following melting of scintillation wax (PerkinElmer MeltiLex A 1450-441) on top of both TCA-precipitated and non-TCA-precipitated filtermats, incorporated radioactivity was measured by a Microbeta2 (PerkinElmer) scintillation counter. Low levels of background radioactivity in S12 and S30 extracts were measured in CFGpS reactions containing no plasmid DNA template and subtracted before calculation of protein yields. The fraction of incorporated leucine (washed/unwashed counts) was multiplied by the overall leucine concentration in the reaction and the molecular weight of the protein, then divided by the number of leucines per protein to determine the amount of protein produced in each reaction. Yields were plotted using Prism v.9.0.0 (GraphPad).

**Autoradiograms of CFGpS reaction products.** For sfGFP-based glycosylation experiments, western blotting of the acceptor proteins followed by densitometry analysis was used to quantitate the fraction of acceptor protein glycosylated (see above for detailed description). For other acceptor proteins, autoradiograms were used to quantitate glycoprotein from CFGpS reaction products using densitometry. Autoradiograms were run by first running SDS-PAGE gels of the soluble fractions of CFGpS reactions (from the same reactions used to calculate yields) using NuPAGE 4–12% Bis-Tris protein gels with MOPS-SDS buffer (Thermo Fisher Scientific, Waltham, MA, USA). The gels were then dried overnight between cellophane films and then exposed for 48–72 h to a Storage Phosphor Screen (GE Healthcare). The Phosphor Screen was imaged using a Typhoon FLA7000 imager (GE Healthcare). Autoradiogram gel images were acquired using Typhoon FLA 7000 Control Software Version 1.2 Build 1.2.1.93. Autoradiogram analysis was performed using ImageJ (Version 2.1.0/1.53c, Build 5f23140693) gel analyzer to determine ratios of glycosylated and aglycosylated full-length acceptor protein. Glycoprotein yields were determined by multiplying fraction glycosylated as determined by ImageJ analysis, by the yields determined from scintillation counting for each replicate. Yields were plotted using Prism v.9.0.0 (GraphPad).

**Cell-free glycoprotein synthesis (CFGpS) reactions.** For crude extract-based expression of glycoproteins, a two-phase scheme was implemented as previously described[34]. In this work, protein synthesis was carried out as described above at 15 μL in PCR strip tubes (Thermo Scientific AB-2000) with 50 ng template DNA. Reactions were supplemented with the plasmids encoding permissible or non-permissible sequons on sfGFP acceptor proteins. pJL1-sfGFP-DQNAT-His (permissible) and pJL1-sfGFP-AQNAT-His (non-permissible) were used for PglB-mediated glycosylation; pJL1-sfGFP-MOOR-His (permissible) and pJL1-sfGFP-MOORmut-His (non-permissible) were used for PglO-mediated glycosylation (Supplementary Table 2). Reactions were set up in triplicate on ice, with each reaction containing a uniquely prepared extract. CFPS time was measured as the time at which reactions were moved to 30 °C to the time when reactions were spiked with MnCl2. In the second phase, protein glycosylation was initiated by the addition of MnCl2 at a final concentration of 25 mM. In addition to MnCl2 (Sigma 63535), either 0.1% (wt/vol.%) DDM (Anatrace, D310S) or 100 mM sucrose was supplemented to PglB or PglO reactions, respectively. Glycosylation proceeded at 30 °C for 16 h. After glycosylation, GFP fluorescence was used to quantitate the

total amount of acceptor protein synthesized, and western blots were used to calculate the fraction of glycosylated and aglycosylated proteins. For additional acceptor proteins shown in Fig. 5f, all CFGpS reaction conditions were held constant other than the DNA template and addition of 10 μM $^{14}$C-leucine to enable quantification (described above).

**Estimation of vesicle membrane area**. Equation 1 below was used to calculate $S$, the vesicle surface area (m$^2$/mL), where $R_{ave}$ is average vesicle radius (m), $C$ is concentration of particles measured by NTA (particles/mL).

$$S = 4*\pi*(R_{ave})^2*C \qquad (1)$$

**Liquid chromatography mass spectrometry (LC-MS/MS)**. Acceptor proteins were purified using a His purification protocol prior to LC-MS. CFGpS reactions producing glycosylated sfGFP-DQNAT and sfGFP-MOOR were scaled up to a total volume of 1.2 mL each and run in 50 mL conical tubes (Falcon, Corning) with 20 min CFPS times. Following 16 h glycosylation reactions, CFGpS reactions were transferred to 1.5 mL microtubes (Axygen, Corning) and centrifuged at 16,000g for 3 min. Soluble fractions were split in half and loaded onto two equilibrated Ni-NTA Spin Columns (Qiagen 31014) per CFGpS reaction following column equilibration with equilibration buffer (50 mM NaH$_2$PO$_4$, 300 mM NaCl and 10 mM imidazole). CFGpS reactions were incubated on columns for 5 min at room temperature followed by centrifugation at 250g for 12 min. Columns were then washed 3 times with 600 μL low-imidazole buffer (50 mM NaH$_2$PO$_4$ and 300 mM NaCl and 20 mM imidazole) and centrifuged at 900g for 2 min before elution in 100 μL of high-imidazole buffer (50 mM NaH$_2$PO$_4$ and 300 mM NaCl and 500 mM imidazole). Four elution fractions were collected, and the most concentrated fraction collected from each column was dialyzed against 50 mM Ammonium Bicarbonate. Dialysis buffer was changed after 2 h and then allowed to proceed overnight.

Glycopeptides for LC-MS/MS analysis were prepared by reducing His-tag purified, dialyzed glycoproteins by incubation with 5 mM DTT at 60 °C for 1 h and then digesting with 0.0044 μg/μL MS Grade Trypsin (Thermo Fisher Scientific, Waltham, MA, USA) at 37 °C overnight. LC-MS/MS was performed by injection of 20 μL (or about 35 pmol for sfGFP-DQNAT and 25 pmol for sfGFP-MOOR) of digested glycopeptides into a Bruker Elute UPLC equipped with an ACQUITY UPLC Peptide BEH C18 Column, 300 Å, 1.7 μm, 2.1 mm × 100 mm (186003686 Waters Corp.) with a 10 mm guard column of identical packing (186004629 Waters Corp.) coupled to an Impact-II UHR TOF Mass Spectrometer. Liquid chromatography was performed using 100% H$_2$O and 0.1% formic acid as Solvent A and 100% acetonitrile and 0.1% formic acid as Solvent B at a flow rate of 0.5 mL/min and a 40 °C column temperature. An initial condition of 0% B was held for 1 min before elution of the peptides of interest during a 4 min gradient to 50% B. The column was washed and equilibrated by a 0.1 min gradient to 100% B, a 2 min wash at 100% B, a 0.1 min gradient to 0% B, and then a 1.8 min hold at 0% B, giving a total 9 min run time[7]. Pseudo multiple reaction monitoring (MRM) MS/MS fragmentation was targeted to theoretical glycopeptide masses corresponding to detected peptide MS peaks. Glycopeptides were fragmented with a collisional energy of 30 eV and an isolation window that included the entire glycopeptide isotopic envelope. For LC-MS/MS of glycopeptides, a scan range of 100–3000 m/z with a spectral rate of 8 Hz was used. External calibration was performed prior to data collection.

LC-MS(/MS) data were collected using Bruker Compass Hystar v5.0 and analyzed using Bruker Compass Data Analysis v4.4 (Bruker Daltonics, Inc.). Representative LC-MS/MS spectra from MRM fragmentation were selected and annotated manually. Observed glycan and glycopeptide m/z values are annotated in figures. LC-MS/MS data were exported from Bruker Compass Data Analysis and plotted in Microsoft Excel.

**Reporting summary**. Further information on research design is available in the Nature Research Reporting Summary linked to this article.

## Data availability

Uniprot accession codes used in this study are: Q5HTX9; Q5FA54; P0AFA2; Q9F7P4; P21554; E9AET9.

All data generated or analyzed during this study are included in the manuscript or Supplementary information or are available from the corresponding author upon request. Accession codes are listed for applicable genes in the Supplementary information. We report no restrictions on data availability. Source data are provided with this paper.

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

## Acknowledgements

We acknowledge Jessica Stark and Weston Kightlinger for helpful discussions about CFGpS, Aravind Natarajan and Dominic Mills for helpful discussions about *O*-OSTs and their substrate specificities, Ashty Karim for helpful discussions on data visualization, Han Teng Wong and Charlotte Abrahamson for fruitful discussions regarding immunostaining, and Maddie DeWinter for helpful discussions on LC-MS/MS methods. We thank Markus Aebi for providing the hR6 serum and Jeff Tabor for providing the DNA encoding NarX. This work made use of the Mass Spectrometry facility of the Integrated Molecular Structure Education and Research Center (IMSERC) at Northwestern University and the BioCryo facility of Northwestern University's NUANCE Center, which has received support from the SHyNE Resource (NSF ECCS-1542205), the IIN, and Northwestern's MRSEC program (NSF DMR-1720139). We gratefully acknowledge support from the Defense Threat Reduction Agency Grant HDTRA1-15-10052/P00001, the National Science Foundation Grants 1936789 and 1844336, the Air Force Research Laboratory Center of Excellence Grant FA8650-15-2-5518, the Bill & Melinda Gates Foundation Grant OPP1217652, the David and Lucile Packard Foundation, and the Camille Dreyfus Teacher-Scholar Program. This project was also supported in part by fellowships awarded to J.M.H. (NDSEG-36373) and K.F.W. (ND-CEN-013-096) through the National Defense Science and Engineering (NDSEG) Fellowship Program, sponsored by the Air Force Research Laboratory, the Office of Naval Research, and the Army Research Office. J.M.H and J.A.P. thank the Ryan Fellowship awarded by Northwestern University. J.A.P. was supported by an NSF Graduate Research Fellowship. The U.S. Government is authorized to reproduce and distribute reprints for Governmental purposes notwithstanding any copyright notation thereon. The views and conclusions contained herein are those of the authors and should not be interpreted as necessarily representing the official policies or endorsements, either expressed or implied, of the Defense Threat Reduction Agency, or the U.S. Government.

## Author contributions

All of the authors designed research; J.M.H., K.F.W., J.A.P., and S.M.I. performed research; C.J.S., J.M.H., J.A.P., and E.W.R. contributed new reagents/analytic tools; J.M.H. and K.F.W. analyzed data; and J.M.H., K.F.W., and M.C.J. wrote the paper. All authors reviewed and edited the paper. M.C.J. provided supervision.

## Competing interests

M.C.J. has a financial interest in Design Pharmaceuticals Inc. and SwiftScale Biologics. M.C.J.'s interests are reviewed and managed by Northwestern University in accordance with their conflict of interest policies. M.P.D. has a financial interest in Glycobia, Inc., Versatope, Inc., Ajuta Therapeutics, Inc., and SwiftScale Biologics. M.P.D.'s interests are reviewed and managed by Cornell University in accordance with their conflict of interest policies. The authors declare no other competing interests.
