## [Peer Review File · Nature Communications]

Reviewers' Comments:

Reviewer #1:

Remarks to the Author:

General aspects:

The Lysate preparation by sonication and homogenization and the production of S12 and S30 extracts was already described in 2015. The addition of extracts with overexpressed PgIB and LLO for the glycosylation of proteins was already described in 2018 (Jaroentomeechai, T., Stark, J.C., Natarajan, A. et al. Single-pot glycoprotein biosynthesis using a cell-free transcription-translation system enriched with glycosylation machinery. *Nat Commun* 9, 2686 (2018). <https://doi.org/10.1038/s41467-018-05110-x>)

Besides the presented production of GFP, which is definitely not a glycoprotein by nature, have other glycosylated proteins been investigated? Can the conditions be transferred to other proteins or do the conditions have to be re-evaluated for each individual target protein?

Detailed aspects:

Page 3, Line 17-20: References 12 and 13 are identical.

Page 4, Line 8: (i) quickly synthesize grams of protein per liter in batch reactions. The references are only showing protein synthesis of grams of sfGFP, but there is no references where something else than a model protein was synthesized in the mentioned range.

Page 6, first section: The characterization of inverted vesicles is not new. In deed inverted membrane vesicles were already characterized 50 years ago. By using a French Press, similar diameters (40-110 nm) were determined for the inverted vesicles. (Reference: „Orientation of Membrane Vesicles from Escherichia coli as Detected by Freeze-Cleave Electron Microscopy“, Altendorf and Staehelin 1973). In addition a publication is available combining a S30 extract for in vitro synthesis with inverted membrane vesicles. „In Vitro Protein Translocation into Escherichia coli Inverted Membrane Vesicles“ Tai et al. 1991)

Page 7, lines 18 – 20; page 9, lines 3 - 4 : It seems obvious, that vesicle concentration increases when centrifugation speed decreases. As a consequence the concentration of the over expressed heterologous cargo membrane proteins should also increase. Why was only S12 examined for vesicle size and not other centrifugation speeds?

Page 8, „Our observations that lysis method impacts vesicle size is consistent with studies showing that varying experimental parameters to disperse phospholipids (or amphiphiles in general) impacts vesicle sizes“. The author is stating by himself that this part lacks novelty.

Page 9, Line 3: „We observed 2-fold membrane protein enrichment in S12 over S30 (S12/S30) extracts for all proteins other than PR, for which we observed 4-fold enrichment „. For PgIO and STT3 a 2 fold enrichment might be fine, but for the other proteins it looks more like a 1.5 fold enrichment.

Page 9, lines 9 - 10: It is not clear how the protein yields ($\mu\text{g/mL}$) were determined and calculated. The authors state that concentrations of (overexpressed) proteins were determined using quantitative western blotting, but the description of the method does not display the necessary Information. In particular the calculation of the values presented in Figure 4, Suppl. Figure 3 and Suppl. Figure 8 E is unclear. Determination of protein yield should be performed more precisely. Incorporation of ^{14}C -labeled amino acids followed by liquid scintillation counting is highly recommended.

Page 9, line 16: „The characteristic vesicle elution peak corresponded with green fluorescence for extracts containing PglB or PglO and no corresponding peak was observed in an extract with no overexpressed membrane protein.“ There is a significant difference in the intensity of the fluorescence Signal detected in both samples. What is the reason for this finding?

Page 10, line 9: „Fluorescence staining and SEC analysis confirmed the presence of LLO and PglB in vesicles (Supplementary Fig. 7A)“. LLO is a lipid-linked oligosaccharide. Both SEC analysis (with and without PglB/PglO) show that LLO is eluting with the vesicles. This result shows that LLO is sticking to the vesicles in an unspecific way.

Page 10, lines 11 – 12: Running the reaction in two phases seems to be necessary due to the addition of the OST cofactor Mn²⁺
Glycosylation and protein folding are processes that also occur co-translationally. It should be discussed which consequence this might have for the synthesis of more complex target proteins besides sfGFP.

Page 10, lines 15 – 17: The authors exclusively use sfGFP as target protein to show successful protein synthesis and also glycosylation by adding PglB/ PglO recognition sites to the sequence. It would be more convincing to see the performance of the established cell-free system for synthesizing naturally existing glycoproteins rather than artificial model proteins.
As the cell-free system described in this manuscript contains an enriched fraction of vesicles after optimization, the system should be able to allow for the synthesis of membrane proteins and should allow translocation and insertion of membrane proteins into the vesicular membrane. As this would really be an important and beneficial additional feature of the system, the authors should further characterize their systems in terms of membrane protein synthesis and translocation.

Page 10, line 26: The protein synthesis determination was performed by Western blot. This is a rough estimation of protein yields lacking the required accuracy and precision. Nowadays protein yields can be calculated with decimal places.
What exactly is the reference that was used for quantification? Why was the protein yield not measured by radioactive scintillation measurement as described earlier in 2015? This method is by far more accurate.

Page 11, Figure 4. The ratio of the bands in the western blot are not congruent with the calculated protein yields. The ratio of g1 should be the same for the different western blots. Have a look to your supplementary figure 10. There the ratios are fine. You have probably chosen a wrong band in figure 4.

Additional proofs of glycosylation, such as enzymatic digestion, are urgently requested.

Page 11, line 18: The authors discriminate between N- and O-glycosylation based on the sequence, but they do not analyze the resulting glycosylation on the protein. As a result the authors can neither be sure about the type of glycosylation nor the exact composition of the sugar moieties. As the authors emphasize the ability of the system to synthesize glycosylated proteins, they should prove which type of glycosylation they detect, e.g. by mass spectrometry or in-gel analysis after digestion with specific glycosidases

Page 12, line 24: Too many vesicles have an inhibitory effect on protein synthesis. This effect may be based on e.g. contaminating proteases or RNases present on the vesicles. The authors should analyse in detail the reason for this inhibitory effect and in Addition it should be analysed at which concentration this effect occurs? Is a ribosome carry-over observed when transferring vesicles into the Lysate?

Page 12, Discussion: N- and O-glycosylation in E. coli (with engineered C. jejuni / Neisseria

glycosylation pathway) results in different glycosylation pattern compared to eukaryotes. The authors should discuss this with respect to possible applications of proteins produced in their system.

Figure 1: In Cryo-EM vesicles are enclosed in larger formations after SEC (Fig. 1D). Please specify in Detail "larger formations". Is it known what this is? It is well known and it was already published that unilaminar and multilaminar vesicles are present in crude extracts. Is the origin of the multilaminar vesicles known? According to the SEC, only unilaminar vesicles are visible in the purified fraction. Were the multilaminar vesicles removed or reshaped by SEC? The arrows in panel C shall indicate vesicles with unilaminar or multilaminar morphology. Unfortunately, this can hardly be seen on the picture. For better comprehensibility, it would be nice to mark the vesicles from the cropped images in the overview image so that the reader can assign them (1C as well 1D).

Reviewer #2:

Remarks to the Author:

This manuscript by Hershewe, Warfel and colleagues describes the important, but little known, role for membrane vesicles in cell-free protein expression systems. The work begins with the use of dynamic light scattering to establish the presence of discrete particle populations in cell-free reactions. The authors go on to characterize the nature of these particles, finding that the population between 100 nm - 200 nm is comprised of lipid vesicles derived from the membranes of E.coli. Using an array of orthogonal approaches (Cryo-EM, DLS, NTA, SEC), work goes on to further confirm the nature of the vesicles in cell-free protein expression systems and explores the effect of upstream lysate preparation protocols on vesicle abundance. The result is a detailed characterization of the vesicles and demonstration of how these vesicles can be used to host functional, molecular payloads that bring new or augmented function to cell-free protein expression systems. This includes the demonstrated delivery of six membrane-bound proteins using engineered vesicles. Perhaps the most exciting is the demonstration of glycosylation, including the first demo of O-linked glycosylation, and the production of glycoproteins with yields exceeding 100 ug/mL.

The manuscript represents a challenging and significant contribution to the emerging field of cell-free synthetic biology and biotechnology. In doing so, authors contribute both a fundamental understanding of the nano-scale membrane vesicles in CFE systems and provide the research community with new levers to begin exploring what has largely been an under appreciated feature of cell-free systems. While I do have minor concerns/comments outlined below, the simplicity of the methods, compelling evidence and potential for translation makes this a strong paper of broad interest.

Comments/concerns:

- A broader audience could probably be reached with a little more context for readers on the importance of glycosylation in protein-based drugs, etc. Why is glycosylation a key consideration for vaccines and other therapeutics?
- Related to this, an opening schematic that maps out the paper would be helpful. E.g. The generation of lipid vesicles from bacterial membranes and the potential for modular loading of protein cargo to membranes, followed by indication of the functional applications.
- Also related to the background information. Some of the glycosylation terminology serves as a barrier to understanding and could be simplified for a general biomedical audience. Similarly, use of the abbreviation "PSD" left me searching for a definition and could probably be left as unabbreviated text.

- Fig. 1A. The use of a grayscale-translucent overlap isn't as clear as it could be. The color regime used in Fig. 4S, at least to me, was much clearer.
- Related to Fig. 1, the data in supplementary Fig 2 B provided a nice level of precision and could be considered for the primary Fig. 1.
- Page 6, line 5. "The 20 nm peak represents small cell-derived particles, including assembled 20 nm E. coli ribosomes⁶², which we confirmed to be active in our extracts (Supplementary Fig. 1)." It is unclear what is being tested here – is this an isolated fraction containing only the 20 nm peak? The generation of GFP from a crude lysate that contains 20 nm diameter particles is likely the result of ribosomes, but I'm not convinced this experiment demonstrates a direct link. This demonstration could be excluded or should be re-written to more clearly present the data and rationale.
- While discussed in the next section, the method of cell lysis for the lysate evaluated in Figure 1 should be mentioned ahead of presentation of the data.
- Figure 3. Labelling of all western blots with S12 and S30 headings would be clearer.
- Figure 4A. Semi-quantitative Western blotting should be mentioned in the legend of Figure 4 as the method used to measure glycosylated protein.
- The method for glycoprotein quantification (ug/mL) in Figure 4 is not clear. How were the units of ug/mL calculated from band intensities? Were titrations of known standards for glycosylated and non-glycosylated forms of GFP evaluated using Western blot to calibrate measurements? Antibody binding to glycosylated protein and non-glycosylated protein is also not necessarily equivalent.
- The purpose of the insets in Fig. 4B,D containing the star symbols is not clear as it is currently presented. Could this information just be placed in the text? These insets currently look like legends for the data, which is confusing.
- Overall, excellent work and presentation of an exciting new aspect of cell-free systems!

RESPONSE TO REVIEWERS

Reviewer #1 (Remarks to the Author):

General aspects:

The Lysate preparation by sonication and homogenization and the production of S12 and S30 extracts was already described in 2015. The addition of extracts with overexpressed PglB and LLO for the glycosylation of proteins was already described in 2018 (Jaroentomeechai, T., Stark, J.C., Natarajan, A. et al. Single-pot glycoprotein biosynthesis using a cell-free transcription-translation system enriched with glycosylation machinery. Nat Commun 9, 2686 (2018). <https://doi.org/10.1038/s41467-018-05110-x>)

We thank the reviewer for providing this opportunity for clarification. The reviewer is correct that our past work (Jaroentomeechai, T., et al., 2018; <https://doi.org/10.1038/s41467-018-05110-x>) described glycosylation-competent *E. coli*-based S30 extracts. However, the use of S12 extracts has not been previously shown for cell-free glycoprotein synthesis. Here, by characterizing and optimizing vesicles for improved glycosylation activity, we show, for the first time, that S12 extracts can be made glyco-competent and importantly enable increased glycoprotein yields as compared to the previous S30 extract-based system.

To address the reviewers concern, we have added clarifying statements in the introduction, results, and discussion of our revised manuscript to better describe our innovations with respect to what's been done previously. Edits to the text:

- *Unfortunately, the existing CFGpS system based on S30 extracts (i.e., cell extracts that result from a 30,000 x g clarification spin) is limited by glycosylation efficiency, only producing ~10-20 µg/mL of glycoprotein in batch.^{2,3,4}*
- *Before this work, S12 extracts had not previously been used for making glyco-competent CFE extracts.*
- *To our knowledge, this is the first time that batch glycoprotein titers on the order of hundreds of µg/mL have been synthesized in a crude-extract-based CFGpS system without extra vesicle supplementation to the reactions². This advance was enabled by using S12 extracts instead of S30 extracts, and relying on enriched cell-derived membranes.*
- *Furthermore, unlike the previously used S30 extract procedure, the optimized S12 extract strategy developed here does not require a high-speed centrifuge and is less time-intensive. This simplifies the CFGpS platform, enabling the process from inoculation of cell culture to testing CFGpS reactions to be completed in a single workday.*

Besides the presented production of GFP, which is definitely not a glycoprotein by nature, have other glycosylated proteins been investigated? Can the conditions be transferred to other proteins or do the conditions have to be re-evaluated for each individual target protein?

We appreciate the reviewer's comment and agree it is important to transfer the conditions to more proteins. In the revised manuscript, we have added new experiments to address this point. Specifically, we now show improvements in glycoprotein titer using S12 extracts (when compared to the base case S30 extracts) using 1 native glycoprotein and 2 conjugate vaccine

carrier proteins. Thus, the reaction conditions can indeed be transferred from GFP to other proteins. Please see below for additions to the text and to **Figure 5**:

*To determine whether enhanced glycoprotein production in S12 extract-based CFGpS reactions was transferrable to non-model acceptor proteins, we tested three additional proteins. This included the Campylobacter jejuni AcrA, a native bacterial glycoprotein with two internal glycosylation sites^{50,84}, as well as two possible carrier proteins for conjugate vaccines. The possible carrier proteins were: H. influenzae protein D (PD), which is a licensed carrier protein, and E. coli maltose binding protein (MBP), which is not yet a licensed carrier but has shown promising results in clinical studies^{85,86}. PD and MBP were each fused to a single C-terminal DQNAT sequon to enable glycosylation²⁴. CFGpS was run as previously described using a 20 min CFPS time. All conditions were held constant other than the DNA template and addition of ¹⁴C-leucine to enable quantification. We observed 80%, 147%, and 167% increases in glycoprotein titers for AcrA, MBP, and PD, respectively, when comparing S12- to S30-based reactions. Expression improvements were determined by ¹⁴C-leucine incorporation (**Fig. 5F, Supplementary Fig. 13A-13E**). Our results highlight that the improvements in glycoprotein yield observed in extracts with higher concentrations of vesicles hold for diverse proteins without the need for optimization.*

See below for the new Figure 5.

Figure 5. Increasing vesicle concentrations improves cell-free glycoprotein synthesis (CFGpS) for N- and O-linked glycosylation systems. For panels **A-E** a standard curve correlating protein yields derived from ^{14}C -Leucine counting and sfGFP fluorescence was used to measure total protein concentrations. Quantitative Western blotting was used to measure fraction of glycosylated protein. For panel **F**, protein concentrations were measured using ^{14}C -Leucine incorporation. Fraction of glycosylated protein was measured using autoradiography. **(A)** sfGFP glycoprotein yields of CFGpS reactions charged with S12 or S30 extracts enriched with PgIB and *C. jejuni* LLO. Error bars represent standard deviation of 3 independent CFGpS reactions, each run with an independent extract. **(Inset)** Schematic of 2-phase CFGpS reactions. **(B)** Glycosylated (dark green) and total (light green) sfGFP yields of N-linked CFGpS reactions with 20-min CFPS times. Error bars represent standard deviation of 3 independent reactions. **(C)** Western blots of acceptor proteins from representative reactions in **(B)**. **(D)** Glycosylated (dark purple) and total (light light) sfGFP yields from O-linked CFGpS reactions with 20 min CFPS times. **(E)** Western blots of acceptor proteins from representative reactions in **(D)**. **(F)** Glycoprotein yields of AcrA, MBP, and PD produced in CFGpS reactions charged with S12 or S30 extracts enriched with PgIB and *C. jejuni* LLO. Glycoprotein yields of AcrA, which contains 2 internal glycosylation sites, include singly and doubly glycosylated protein. Error bars represent standard deviation of 3 independent CFGpS reactions, each run with an independent extract.

Detailed aspects:

Page 3, Line 17-20: References 12 and 13 are identical.

Thank you for pointing out this reference duplication. It has been corrected.

Page 4, Line 8: (i) quickly synthesize grams of protein per liter in batch reactions. The references are only showing protein synthesis of grams of sfGFP, but there is no references where something else than a model protein was synthesized in the mentioned range.

We agree with the reviewer. To address this point, we added additional references describing batch CFPS of ~20 non-model enzymes (Kightlinger, W., *et al.*) and ~15 biotherapeutic proteins such as antibodies and GM-CSF (Cai *et al.*). These two citations describe proteins synthesized in *E. coli* systems to titers on the order of grams/L protein. Specifically, we write:

Optimized E. coli-based CFE reactions: (i) quickly synthesize grams of protein per liter in batch reactions^{14–18}, [...] With the new references being:

17. Kightlinger, W. *et al.* A cell-free biosynthesis platform for modular construction of protein glycosylation pathways. *Nat. Commun.* **10**, 1–13 (2019).
18. Cai, Q. *et al.* A simplified and robust protocol for immunoglobulin expression in *Escherichia coli* cell-free protein synthesis systems. *Biotechnol. Prog.* **31**, 823–831 (2015).

Page 6, first section: The characterization of inverted vesicles is not new. In deed inverted membrane vesicles were already characterized 50 years ago. By using a French Press, similar diameters (40-110 nm) were determined for the inverted vesicles. (Reference: „Orientation of Membrane Vesicles from *Escherichia coli* as Detected by Freeze-Cleave Electron Microscopy“, Altendorf and Staehelin 1973). In addition a publication is available combining a S30 extract for in vitro synthesis with inverted membrane vesicles. „In Vitro Protein Translocation into *Escherichia coli* Inverted Membrane Vesicles“ Tai *et al.* 1991)

We agree that characterization of inverted vesicles has a long history and thank the reviewer for drawing our attention to 2 additional citations that we did not include in our original manuscript. In the revised manuscript, these references have been added. Throughout the text, we also now better emphasize previous literature pertaining to characterization of 1) *E. coli*-derived vesicles in CFE extracts, 2) deconstructed systems with purified lipids, 3) CFE systems with membrane fraction supplementation (like the second reference mentioned above), and 4) lysed *E. coli* cells (like the first reference mentioned above). The vesicles observed in our crude extracts are consistent with previous works.

While characterizing vesicles in and of itself is not new, we appreciate the reviewer for giving us the opportunity to clarify what is new in this study; namely, (i) characterizing vesicles in crude extracts for bacterial glycoengineering, including showing that both LLO and OST are associated with vesicles, (ii) showing the impact of increasing vesicle content to increase glycoprotein yields, and (iii) applying facile characterization techniques such as light scattering to directly analyze vesicles in extracts without the need for lengthy TEM protocols. This is now better articulated in the revised manuscript.

Edits to text include:

- *Despite the absence of intact cellular membranes, membrane structures are present in crude extract-based CFE systems. They form upon fragmentation and rearrangement of cell membranes during cell lysis and extract preparation and have been studied and characterized for decades^{51,54-57}.*
- *For example, membrane augmented cell-free systems achieved through the use of nanodiscs, synthetic phospholipid structures, purified microsomes, and purified vesicles have all enabled the use of membrane components in CFE systems⁴⁵⁻⁵¹.*
- *To do so, we apply canonical strategies (e.g., TEM), but also apply simple and expedited characterization workflows that rely on techniques such as light scattering to directly analyze vesicles in extracts without the need for lengthy protocols.*

Page 7, lines 18 – 20; page 9, lines 3 - 4 : It seems obvious, that vesicle concentration increases when centrifugation speed decreases. As a consequence the concentration of the over expressed heterologous cargo membrane proteins should also increase. Why was only S12 examined for vesicle size and not other centrifugation speeds?

We focused on S12-based extracts and not others because there is rich literature on developing highly active cell-free protein synthesis systems at this centrifugation speed. As a consequence, we did not want to change this variable so we could build off those works. In the revised manuscript, we added the sentence to the results section explicitly stating our reason for choosing these speeds in the text below. We state:

*These combinations of lysis and centrifugation protocols were selected because they have previously been used to obtain high-yielding *E. coli* CFE extracts⁷⁸. Indeed, all the conditions tested yielded extracts that were active for protein synthesis in standard CFE reaction conditions (Supplementary Fig. 4A).*

Additionally, we added a line to explain this result more directly:

We observed significantly higher numbers of vesicles in S12 extracts for both lysis methods, with the reduced centrifugation speed likely being the reason for increased particle concentrations.

That said, we agree that testing other speeds is a variable for tuning vesicle size and have additionally included this in the revised text.

Importantly, our workflow easily interfaces with numerous methods that could be used to alter vesicles and their membrane-bound cargo. For example, using other centrifugation speeds besides 12,000 x g could result in changes to vesicle concentration. In addition, additives could be added to cell-free systems to tune biophysical features of membrane properties (e.g., composition, size, fluidity, curvature).

Page 8, „Our observations that lysis method impacts vesicle size is consistent with studies showing that varying experimental parameters to disperse phospholipids (or amphiphiles in general) impacts vesicle sizes“. The author is stating by himself that this part lacks novelty.

Our goal with this statement is to help explain why vesicle sizes differ between the two cell-extract preparation methods. Work with synthetic lipids, for example, have shown that different methods of lipid dispersion (e.g., probe sonication vs. gentle agitation) can have significant

effects on vesicle sizes and those effects are likely at play in this study. In addition, we are not the first to study vesicles in crude extracts, nor do we claim to be. However, we believe this study is novel in several ways with respect to vesicle characterization. Namely, we were the first to our knowledge to (i) characterize vesicles in crude extracts for glycoengineering, including showing that both LLO and OST are associated with vesicles, (ii) show the impact of increasing vesicle content to increase glycoprotein yields, and (iii) apply facile characterization techniques such as light scattering to directly analyze vesicles in extracts without the need for lengthy TEM protocols. Importantly, these innovations led to technological improvements for cell-free glycoprotein synthesis that we show in the revised manuscript can be applied to multiple proteins.

We now add the following text to the discussion:

By applying our optimized methods to increase concentrations of vesicle-bound glycosylation machinery, we shorten the time associated with extract preparation, increase glycosylation efficiencies, and enhance glycoprotein titers by up to ~170%. Importantly, we go on to show that improvements in glycoprotein titers are generalizable to multiple glycoproteins without the need to re-optimize conditions.

Page 9, Line 3: „We observed 2-fold membrane protein enrichment in S12 over S30 (S12/S30) extracts for all proteins other than PR, for which we observed 4-fold enrichment „. For PglO and STT3 a 2 fold enrichment might be fine, but for the other proteins it looks more like a 1.5 fold enrichment.

We thank the reviewer for raising this point of clarification and agree it is difficult to tell from the Western blots. To clarify, we have now added the raw data (rounded to the hundredths place) used to calculate S12/S30 enrichment from the densitometry analysis to **Supplemental Figure 6**. Reported standard deviations <0.01 are reported as 0.01 in the table. Data in **Figure 4** align with the newly reported data for S12/S30 enrichment and are now reported rounded to the nearest tenths place.

Protein	S12 avg densitometry signal	S12 std dev densitometry signal	S30 avg densitometry signal	S30 std dev densitometry signal	S12/S30 densitometry signal	S12/S30 error densitometry signal	S12/S30 ratio rounded for Figure
NarX	2.76	0.70	1.44	0.37	1.91	0.69	1.9
PR	0.83	0.22	0.21	0.03	3.88	1.16	3.9
CB1	0.25	0.01	0.13	0.01	1.88	0.11	1.9
PglO	8.71	0.36	4.66	0.37	1.87	0.17	1.9
PglB	5.23	0.10	3.47	0.18	1.51	0.08	1.5
LmSTT3	0.16	0.01	0.08	0.01	2.14	0.12	2.1
sfGFP	16.87	0.72	16.90	0.26	1.00	0.05	1.0

Page 9, lines 9 - 10: It is not clear how the protein yields ($\mu\text{g/mL}$) were determined and calculated. The authors state that concentrations of (overexpressed) proteins were determined using quantitative western blotting, but the description of the method does not display the necessary information. In particular the calculation of the values presented in Figure 4, Suppl. Figure 3 and Suppl. Figure 8 E is unclear. Determination of protein yield should be performed more precisely. Incorporation of ^{14}C -labeled amino acids followed by liquid scintillation counting is highly recommended.

We see how our methods seemed unclear before and appreciate the suggestion about using incorporation of ^{14}C -labeled amino acids followed by liquid scintillation counting. Importantly, Western blotting was not used to calculate protein concentrations. In the revised manuscript, ^{14}C -labeled leucine followed by liquid scintillation counting was used to quantitate protein concentrations. In the case of sfGFP, we used a standard curve correlating ^{14}C -derived protein concentration with sfGFP fluorescence to calculate concentrations for ease of use. However, ^{14}C -labeled leucine followed by liquid scintillation counting was the basis. The standard curve is shown here for the reviewer's reference. This method was used to determine the concentration of sfGFP reported in the following figures: **Figure 5, Supplementary Figure 4 and Supplementary Figure 9E.**

For clarity, instead of referencing another publication to describe the quantification methods, our methods section has now been updated to describe our ^{14}C -leucine quantitation methods. Specifically, we write:

Quantification of CFE and CFGpS protein yields

As described previously, the concentration of cell-free-derived sfGFP was determined by measuring in-extract fluorescence and then converting to protein concentration using a standard curve relating sfGFP fluorescence to protein concentration as determined by a [^{14}C]-leucine incorporation assay³⁴. Briefly, 2 μL of cell-free reaction product was diluted into 48 μL of Ambion nanopure water (Invitrogen, USA). The solution was then placed in a Costar 96-well black assay plate (Corning, USA). Fluorescence was measured using a Synergy H1 microplate reader (Biotek, USA). Excitation and emission wavelengths for sfGFP fluorescence were 485 and 528 nm, respectively. This RFU value was then used to calculate the protein concentration.

Yields of all acceptor proteins (other than sfGFP) were assessed directly via the addition of 10 μM [^{14}C]-leucine (PerkinElmer) to the CFGpS reaction to yield trichloroacetic acid-precipitable radioactivity that was measured using scintillation counting. Soluble fractions were isolated after centrifugation at $\geq 12,000 \times g$ for 15 min at 4 $^{\circ}\text{C}$. Briefly, 6 μL of the soluble fraction of CFGpS reactions run with 20 min CFPS times were mixed with 6 μL 0.5 M KOH and incubated for 20 min at 37 $^{\circ}\text{C}$. 5 μL of treated sample was then soaked into 2 separate filtermats (PerkinElmer Printer Filtermat A 1450-421) and dried under a heat lamp. One filtermat was washed three times using 5% trichloroacetic acid (TCA) with 15 min incubations at 4 $^{\circ}\text{C}$, and then once with ethanol with a 10 min incubation at room temperature. Following melting of scintillation wax (PerkinElmer MeltiLex A 1450-441) on top of both TCA-precipitated and non-TCA-precipitated filtermats, incorporated radioactivity was measured by a Microbeta2 (PerkinElmer) scintillation counter. Low levels of background radioactivity in S12 and S30 extracts were measured in CFGpS reactions containing no plasmid DNA template and subtracted before calculation of

protein yields. The fraction of incorporated leucine (washed/unwashed) was used to determine the amount of protein produced in each reaction.

Page 9, line 16: „The characteristic vesicle elution peak corresponded with green fluorescence for extracts containing PglB or PglO and no corresponding peak was observed in an extract with no overexpressed membrane protein.“ There is a significant difference in the intensity of the fluorescence signal detected in both samples. What is the reason for this finding?

The peak observed in the extract with no over-expressed membrane protein occurs after the fractions containing vesicles. As this peak indicates antibody not associated with vesicles and occurs in the control lacking protein, we hypothesize that this peak is unbound antibody eluting from the column in later fractions. The difference in fluorescence intensities between samples can be attributed to different protein expression levels between PglB and PglO enriched extracts. This is not meant as a quantitative comparison between lysates but rather confirmation of the presence of each protein in the vesicle fraction.

Page 10, line 9: „Fluorescence staining and SEC analysis confirmed the presence of LLO and PglB in vesicles (Supplementary Fig. 7A)“. LLO is a lipid-linked oligosaccharide. Both SEC analysis (with and without PglB/PglO) show that LLO is eluting with the vesicles. This result shows that LLO is sticking to the vesicles in an unspecific way.

We thank the reviewer for raising this point of clarification. The reviewer may be referring to a small amount of background binding of the SBA lectin to extracts that do not contain LLO. These traces give a baseline for signal and do suggest some level of nonspecific binding, however, samples containing LLO have significantly higher signal than negative controls. To clarify this point, we added the following sentence to the figure caption for Supplementary Fig. 8:

A low amount of nonspecific binding of the α -LLO SBA lectin is observed and serves as a signal baseline for the LLO-containing samples.

Since we did not prove if the LLO is just associated to or physically embedded in the membrane, we also edited this sentence:

*Fluorescence staining and SEC analysis confirmed the presence and association of LLO and PglB with the vesicles (**Supplementary Fig. 8A**).*

Page 10, lines 11 – 12: Running the reaction in two phases seems to be necessary due to the addition of the OST cofactor Mn²⁺. Glycosylation and protein folding are processes that also occur co-translationally. It should be discussed which consequence this might have for the synthesis of more complex target proteins besides sfGFP.

We agree this is an important point to discuss in the manuscript. We now have the following text in the discussion to address this:

Future studies to elucidate translocation and co-translational glycosylation in vesicles will be important. These studies could be especially useful for producing complex, native glycoproteins for which protein glycosylation and folding are co-translational. While it has been shown that glycosylation with PglB can proceed on pre-folded proteins in vitro (using purified, reconstituted components and without the need for translocation or intact membranes⁹¹), obtaining a more robust understanding of the topology of glycosylation in membrane vesicles is an important future effort for therapeutics production.

Page 10, lines 15 – 17: The authors exclusively use sfGFP as target protein to show successful protein synthesis and also glycosylation by adding PglB/ PglO recognition sites to the sequence. It would be more convincing to see the performance of the established cell-free system for synthesizing naturally existing glycoproteins rather than artificial model proteins.

We agree with the reviewer that we should have added non-GFP proteins. In the revised manuscript, we have now added additional non-GFP target proteins to show improvements in glycosylation and successful glycosylation in our system with other proteins (including the native glycoprotein *AcrA* and vaccine carriers – see **Figure 5** and described above). While the cell-free glycosylation of naturally existing glycoproteins is an important goal, the glycosylation of proteins that are not naturally glycosylated is also an important advance, particularly when modifying bacterial proteins for use as vaccines. For example, the use of fusion proteins containing PglB/PglO recognition sequences is a core advance in bacterial glycoengineering for making non-native or engineered proteins such as conjugate vaccines, an application that is particularly well-suited for our system.

As the cell-free system described in this manuscript contains an enriched fraction of vesicles after optimization, the system should be able to allow for the synthesis of membrane proteins and should allow translocation and insertion of membrane proteins into the vesicular membrane. As this would really be an important and beneficial additional feature of the system, the authors should further characterize their systems in terms of membrane protein synthesis and translocation.

We share that reviewer's enthusiasm for characterizing translocation into vesicles in our new cell-free glycoprotein synthesis system. However, our work focuses on the use of bacterial glycosylation systems for glycoengineering based on PglB and LLOs, which do not require translocation or co-translational glycosylation *in vitro*. As such and given the scope of the existing manuscript, we did not pursue this. However, to address the point raised and since we agree with the reviewer that this is an exciting direction, we now highlight this additional opportunity in the discussion of the revised work:

Future studies to elucidate translocation and co-translational glycosylation in vesicles will be important. These studies could be especially useful for producing complex, native glycoproteins for which protein glycosylation and folding are co-translational. While it has been shown that glycosylation with PglB can proceed on pre-folded proteins in vitro (without the need for translocation or intact membranes), obtaining a more robust understanding of the topology of glycosylation in membrane vesicles is an important future effort for therapeutics production.

Page 10, line 26: The protein synthesis determination was performed by Western blot. This is a rough estimation of protein yields lacking the required accuracy and precision. Nowadays protein yields can be calculated with decimal places.

What exactly is the reference that was used for quantification? Why was the protein yield not measured by radioactive scintillation measurement as described earlier in 2015? This method is by far more accurate.

We agree for the need to be more quantitative. In the revised manuscript, ¹⁴C-leucine incorporation was used to quantitate cell-free expression yields. The text and methods have now been updated and clarified. This is also described above.

Page 11, Figure 4. The ratio of the bands in the western blot are not congruent with the calculated protein yields. The ratio of g1 should be the same for the different western blots. Have a look to your supplementary figure 10. There the ratios are fine. You have probably chosen a wrong band in figure 4.

Thank you for pointing this error out. The error arose from a scaling issue that has now been corrected.

Additional proofs of glycosylation, such as enzymatic digestion, are urgently requested.

We agree with the reviewer about the need for additional verification of glycosylation. In the revised manuscript, we have confirmed glycosylation on our model *N*- and *O*-linked sfGFP glycoproteins using LC-MS/MS, which is now available in the supplement (**Supplementary Figure 11**). LC-MS/MS was pursued because no commercially available enzymes to characterize deglycosylation are routinely used with this particular glycan. This data, paired with 1) cross-reactivity of the glycoprotein with a *C. jejuni*-specific rabbit-derived serum on a Western Blot, 2) mass shifts observed on anti-acceptor protein blots, 3) loss of glycosylation activity upon mutation of the Asp (*N*-linked) or Ser (*O*-linked) residue within the sequon, and 4) binding activity of the SBA lectin to LLO-enriched vesicles all provide robust proof of glycosylation and glycan structure.

Edits to text (results):

As additional proof of site-specific glycosylation, we performed LC-MS/MS analysis of the glycoproteins obtained via CFGpS with PglO and PglB and observed the presence of the 1406 Da C. jejuni heptasaccharide on the expected tryptic peptides (Supplementary Figure 11A-11B)⁸⁰.

New supplementary figure:

A

B

Supplementary Figure 11. LC-MS/MS of trypsin digested glycopeptides. LC-MS/MS was performed with a Bruker Elute UPLC coupled to an Impact-II UHR TOF Mass Spectrometer. **(A)** A quadruply-charged precursor ion (denoted with a blue diamond) was identified as the glycopeptide (LISEEDLNGAALGGDQDATGGHHHHHH) digested from sfGFP-DQNAT (predicted *m/z* 1090.5). Fragmentation with an isolation window that included the entire glycopeptide isotopic envelope with 30 eV revealed glycan fragment ions as well as intact peptide with fragmented glycan characteristic of the *C. jejuni* glycan. Highest intensity peaks are labeled and are +1 charge states unless otherwise indicated. **(B)** A triply-charged precursor ion (denoted with a blue diamond) was identified as the glycopeptide (NVGGDLDPAAASAPQPGKPPR) digested from sfGFP-MOOR (predicted *m/z* 1202.9). Fragmentation with an isolation window that included the entire glycopeptide isotopic envelope with 30 eV also revealed characteristic glycan fragment ions and intact peptide with fragmented glycan characteristic of the *C. jejuni* glycan. Highest intensity peaks are labeled and are +1 charge states unless otherwise indicated. Previous reports and glycosylation site amino acid mutation studies shown in **Supplementary Figure 10** strongly suggest that the glycan modification is on the bolded **N** and **S** residues within the sequons on sfGFP-DQNAT and sfGFP-MOOR glycopeptides, respectively.

Edits to methods:

Liquid Chromatography Mass Spectrometry (LC-MS/MS)

Acceptor proteins were purified using a His purification protocol prior to LC-MS. CFGpS reactions producing glycosylated sfGFP-DQNAT and sfGFP-MOOR were scaled up to a total volume of 1.2 mL each and run in 50 mL conical tubes (Falcon, Corning) with 20-minute CFPS times. Following 16-hour glycosylation reactions, CFGpS reactions were transferred to 1.5 mL microtubes (Axygen, Corning) and centrifuged at 16,000 x g for 3 minutes. Soluble fractions were split in half and loaded onto 2 equilibrated Ni-NTA Spin Columns (Qiagen 31014) per CFGpS reaction following column equilibration with equilibration buffer (50 mM NaH₂PO₄, 300 mM NaCl and 10 mM imidazole). CFGpS reactions were incubated on columns for 5 minutes at room temperature followed by centrifugation at 250 x g for 12 minutes. Columns were then washed 3 times with 600 μ L low imidazole buffer (50 mM NaH₂PO₄ and 300 mM NaCl and 20 mM imidazole) and centrifuged at 900 x g for 2 minutes before elution in 100 μ L of high-imidazole buffer (50 mM NaH₂PO₄ and 300 mM NaCl and 500 mM imidazole). Four elution fractions were collected, and the most concentrated fraction collected from each column was dialyzed against 50 mM Ammonium Bicarbonate. Dialysis buffer was changed after 2 hours and then allowed to proceed overnight.

Glycopeptides for LC-MS/MS analysis were prepared by reducing His-tag purified, dialyzed glycoproteins by incubation with 5 mM DTT at 60 °C for 1 hour and then digesting with 0.0044 μ g/ μ L MS Grade Trypsin (Thermo Fisher Scientific) at 37 °C overnight. LC-MS/MS was performed by injection of 20 μ L (or about 35 pmol for sfGFP-DQNAT and 25 pmol for sfGFP-MOOR) of digested glycopeptides into a Bruker Elute UPLC equipped with an ACQUITY UPLC Peptide BEH C18 Column, 300 Å, 1.7 μ m, 2.1 mm x 100 mm (186003686 Waters Corp.) with a 10 mm guard column of identical packing (186004629 Waters Corp.) coupled to an Impact-II UHR TOF Mass Spectrometer. As described previously, liquid chromatography was performed using 100% H₂O and 0.1% formic acid as Solvent A and 100% acetonitrile and 0.1% formic acid as Solvent B at a flow rate of 0.5 mL/min and a 40 °C column temperature. An initial condition of 0% B was held for 1 min before elution of the peptides of interest during a 4 min gradient to 50% B. The column was washed and equilibrated by a 0.1 min gradient to 100% B, a 2 min wash at 100% B, a 0.1 min gradient to 0% B, and then a 1.8 min hold at 0% B, giving a total 9 min run time¹⁷. Pseudo multiple reaction monitoring (MRM) MS/MS fragmentation was targeted to theoretical glycopeptide masses corresponding to detected peptide MS peaks. Glycopeptides were fragmented with a collisional energy of 30 eV and an isolation window that included the entire glycopeptide isotopic envelope. For LC-MS/MS of glycopeptides, a scan range of 100–3000 m/z with a spectral rate of 8 Hz was used. External calibration was performed prior to data collection.

LC-MS(/MS) data was collected using Bruker Compass Hystar v5.0 and analyzed using Bruker Compass Data Analysis v4.4 (Bruker Daltonics, Inc.). Representative LC-MS/MS spectra from MRM fragmentation were selected and annotated manually. Observed glycan and glycopeptide m/z values are annotated in figures. LC-MS/MS data was exported from Bruker Compass Data Analysis and plotted in Microsoft Excel.

Page 11, line 18: The authors discriminate between N- and O-glycosylation based on the sequence, but they do not analyze the resulting glycosylation on the protein. As a result the authors can neither be sure about the type of glycosylation nor the exact composition of the sugar moieties. As the authors emphasize the ability of the system to synthesize glycosylated proteins, they should prove which type of glycosylation they detect, e.g. by mass spectrometry or in-gel analysis after digestion with specific glycosidases

In the revised manuscript, we have added mass spectrometry data on the glycan structure via LC-MS/MS on trypsin digested glycoproteins to discriminate between *N*- and *O*-glycosylation. Please see above. Additionally, we confirmed a loss of glycosylation on acceptor proteins with a single point mutation to render the sequon 'unpermissible' to either *N*- or *O*-glycosyltransferases.

Page 12, line 24: Too many vesicles have an inhibitory effect on protein synthesis. This effect may be based on e.g. contaminating proteases or RNAses present on the vesicles. The authors should analyse in detail the reason for this inhibitory effect and in Addition it should be analysed at which concentration this effect occurs? Is a ribosome carry-over observed when transferring vesicles into the Lysate?

We agree with the reviewer that different extract preparation methods (S12 or S30) can alter cell-free protein synthesis. It is higher in some cases and lower in others. Looking forward, we share the reviewer's interest in understanding causes for expression differences (e.g., vesicles, soluble proteins, energy regeneration, etc.). This can be pursued in a follow-on study.

Page 12, Discussion: *N*- and *O*-glycosylation in *E. coli* (with engineered *C. jejuni* / *Neisseria* glycosylation pathway) results in different glycosylation pattern compared to eukaryotes. The authors should discuss this with respect to possible applications of proteins produced in their system.

We thank the reviewer for this point and now discuss possible applications of proteins with *N*- and *O*-glycosylation. Importantly, we clarify and highlight that the system described here is, at present, most useful for expressing recombinant bacterial glycosylation machinery with an eye toward conjugate vaccine production (especially in comparison to other CFGpS systems derived from eukaryotic cell extracts that contain endogenous eukaryotic glycosylation machinery). Indeed, the expression of eukaryotic glycosylation machinery is possible in *E. coli*, but remains challenging. We have added a discussion of this and the advantages/ limitations of our system in the discussion:

Towards applications in biomanufacturing, a key feature of the E. coli-based CFGpS system is expressing synthetic glycosylation pathways encoding diverse O-antigens from pathogenic bacteria. This feature points toward immediate utility of our CFGpS system in the on-demand bioproduction of conjugate vaccines²⁴. Here, we show that S12 extracts enable higher glycoprotein titers of two glycoconjugate vaccine carrier proteins modified with a model C. jejuni LLO, indicating that vaccine production may be simpler and more efficient using the optimized methods reported here. Additionally, we have recently shown that our optimized S12 conditions can be used to recapitulate efficient, humanized O-linked glycosylation in glycoengineered E. coli extracts⁷². While applications in O-linked glycosylation and conjugate vaccines are imminent, the recapitulation of efficient eukaryotic-type N-linked glycosylation (i.e., glycoproteins with a Man₃GlcNAc₂ core glycan) for therapeutics production still remains on the horizon in E. coli-based systems.

Future studies to elucidate translocation and co-translational glycosylation in vesicles will be important. These studies could be especially useful for producing complex, native glycoproteins for which protein glycosylation and folding are co-translational. While it has been shown that glycosylation with PglB can proceed on pre-folded proteins in vitro (using purified, reconstituted components and without the need for translocation or intact

membranes⁹¹), obtaining a more robust understanding of the topology of glycosylation in membrane vesicles is an important future effort for therapeutics production.

Figure 1: In Cryo-EM vesicles are enclosed in larger formations after SEC (Fig. 1D). Please specify in Detail "larger formations". Is it known what this is? It is well known and it was already published that unilaminar and multilaminar vesicles are present in crude extracts. Is the origin of the multilaminar vesicles known? According to the SEC, only unilaminar vesicles are visible in the purified fraction. Were the multilaminar vesicles removed or reshaped by SEC? The arrows in panel C shall indicate vesicles with unilammellar or multilammellar morphology. Unfortunately, this can hardly be seen on the picture. For better comprehensibility, it would be nice to mark the vesicles from the cropped images in the overview image so that the reader can assign them (1C as well 1D).

We thank the reviewer for these questions which are directly addressed in the revised manuscript. To do so, a supplemental figure showing the origin of cropped vesicles has been added as **Supplementary Figure 3**. This figure also includes an expanded set of TEM images that should help the reader to discern what a typical CryoEM micrograph of extract and purified vesicles looks like. Of note, this figure has also been marked per the reviewer's suggestion.

Regarding 'larger formations': the reviewer may be referring to the TEM grid itself. The larger formations are not vesicles, but the smaller particles inside are. To give more information about size distribution of purified vesicles, we have also added in-solution scattering data (NTA particle size distribution) of purified vesicles to **Figure 2**. As observed in the DLS data, there is a distribution of vesicle sizes with some larger vesicles, greater than 100 nm, present in solution. The SEC resin elutes particles larger than a threshold size in the void volume where vesicles appear, and thus would not be expected to exclude vesicles larger in size. We would also note that, while Cryo-EM is an excellent way to validate size and morphology, it is a small snapshot of the sample, showing a few particles. Thus, EM images should be analyzed together with other scattering methods for the most rigorous analysis.

To the questions regarding multilammellar vesicles: the nesting of vesicles could occur during the process of lysis and rearrangement of the membrane. Based on our Cryo EM results in crude extract, the formation of unilammellar vesicles is more likely than the nested morphology. After SEC, we do observe the presence of some multilammellar vesicles in the TEM. For clarity and to avoid it being difficult to see, we have now highlighted multilammellar vesicles in the cropped images in in **Figure 2**.

Edited Figure 2:

Figure 2. Characterization of membrane vesicles in crude CFE extracts. (A) DLS analysis of crude extracts and SEC purified vesicles. Error bars represent the standard deviation within triplicate analysis of three independently-prepared extracts. For purified vesicles, error bars represent the standard deviation of triplicate analysis of the most concentrated vesicle elution fraction. (B) NTA of purified vesicles collected from SEC. Mean and mode diameters observed in the particle size distribution are listed in the inset. (C) Illustration of particles detected in crude CFE extracts. (D) CryoEM micrographs of crude extracts. Black arrows indicate vesicles with apparent unilamellar morphology. White arrows indicate nested or multilamellar morphologies. Cropped images indicate representative vesicles. Scale bars are 100 nm. Uncropped images are available in Supplementary Fig. 3 and numbered with the corresponding cropped vesicles. (E) CryoEM micrographs of SEC purified vesicles. Cropped images indicate representative purified vesicle particles. Scale bars are 100 nm. Uncropped images are available in Supplementary Fig. 3 and numbered with the corresponding cropped vesicles.

New Supplementary Figure 3:

Supplementary Figure 3. Cryo-EM analysis of vesicles in crude extracts (top) and after SEC purification (bottom). The numbering of vesicles in uncropped images corresponds with the vesicle shown in each cropped image.

Reviewer #2 (Remarks to the Author):

This manuscript by Hershewe, Warfel and colleagues describes the important, but little known, role for membrane vesicles in cell-free protein expression systems. The work begins with the use of dynamic light scattering to establish the presence of discrete particle populations in cell-free reactions. The authors go on to characterize the nature of these particles, finding that the population between 100 nm - 200 nm is comprised of lipid vesicles derived from the membranes of *E. coli*. Using an array of orthogonal approaches (Cryo-EM, DLS, NTA, SEC), work goes on to further confirm the nature of the vesicles in cell-free protein expression systems and explores the effect of upstream lysate preparation protocols on vesicle abundance. The result is a detailed characterization of the vesicles and demonstration of how these vesicles can be used to host functional, molecular payloads that bring new or augmented function to cell-free protein expression systems. This includes the demonstrated delivery of six membrane-bound proteins using engineered vesicles. Perhaps the most exciting is the demonstration of glycosylation, including the first demo of O-linked glycosylation, and the production of glycoproteins with yields exceeding 100 ug/mL.

We thank the reviewer for their insightful comments and support of our work.

The manuscript represents a challenging and significant contribution to the emerging field of cell-free synthetic biology and biotechnology. In doing so, authors contribute both a fundamental understanding of the nano-scale membrane vesicles in CFE systems and provide the research community with new levers to begin exploring what has largely been an under appreciated feature of cell-free systems. While I do have minor concerns/comments outlined below, the simplicity of the methods, compelling evidence and potential for translation makes this a strong paper of broad interest.

We are grateful for the recognition of our work and the acknowledgment of the of broad interest of our methods and findings.

Comments/concerns:

- A broader audience could probably be reached with a little more context for readers on the importance of glycosylation in protein-based drugs, etc. Why is glycosylation a key consideration for vaccines and other therapeutics?

We thank the reviewer for this point. We have expanded the discussion in the text as to why glycosylation is important for vaccines and therapeutics. Specifically, we added:

Introduction:

For example, protein glycosylation, which can profoundly impact folding, stability, and activity of proteins and therapeutics⁶⁶⁻⁶⁹, is mediated by membrane-bound components. Introduction of cell-derived vesicles with machinery required for glycosylation could enable cell-free biomanufacturing of protein therapeutics and conjugate vaccines.

Main text:

With an eye towards bacterial glycoengineering applications, we[...]

We focused on protein glycosylation, because glycosylation plays critical roles in cellular function, human health, and biotechnology.

Discussion:

Towards applications in biomanufacturing, a key feature of the E. coli-based CFGpS system is expressing synthetic glycosylation pathways encoding diverse O-antigens from pathogenic bacteria. This feature points toward immediate utility of our CFGpS system in the on-demand bioproduction of conjugate vaccines²⁴.

- Related to this, an opening schematic that maps out the paper would be helpful. E.g. The generation of lipid vesicles from bacterial membranes and the potential for modular loading of protein cargo to membranes, followed by indication of the functional applications.

Thank you for this useful suggestion. We have now added an opening schematic that maps out the paper. This appears as the new Figure 1 of the revised manuscript. Please see below.

Figure 1. Overview schematic of engineering CFE systems with cell-derived membrane-dependent functions. Membrane-bound cargo expressed in living *E. coli* is carried through into CFE extracts via membrane vesicles. The extract preparation method used to prepare CFE extracts impacts sizes and concentrations of vesicles, and their associated cargo. Here, we developed a facile nanocharacterization pipeline to better understand and characterize the impacts of extract preparation methods on vesicle profiles and their associated cargo. We then apply our findings to improve cell-free glycoprotein synthesis, which is a promising platform for on-demand vaccine development. By increasing concentrations of vesicles and membrane-bound glycosylation machinery (OST and LLO), we overcome limitations in cell-free glycoprotein synthesis and significantly increase glycoprotein titers.

- Also related to the background information. Some of the glycosylation terminology serves as a barrier to understanding and could be simplified for a general biomedical audience. Similarly, use of the abbreviation “PSD” left me searching for a definition and could probably be left as unabbreviated text.

We have expanded our definitions and simplified the glycosylation terminology throughout the abstract/introduction to remove jargon. We removed the abbreviation PSD.

- Fig. 1A. The use of a grayscale-translucent overlap isn’t as clear as it could be. The color regime used in Fig. 4S, at least to me, was much clearer.

We changed the grayscale translucent overlap in **Figure 2A** to the blue and green color scheme to show the data more clearly (see below).

Figure 2. Characterization of membrane vesicles in crude CFE extracts. (A) DLS analysis of crude extracts and SEC purified vesicles. Error bars represent the standard deviation within triplicate analysis of three independently-prepared extracts. For purified vesicles, error bars represent the standard deviation of triplicate analysis of the most concentrated vesicle elution fraction. (B) NTA of purified vesicles collected from SEC. Mean and mode diameters observed in the particle size distribution are listed in the inset. (C) Illustration of particles detected in crude CFE extracts. (D) CryoEM micrographs of crude extracts. Black arrows indicate vesicles with apparent unilamellar morphology. White arrows indicate nested or multilamellar morphologies. Cropped images indicate representative vesicles. Scale bars are 100 nm. Uncropped images are available in **Supplementary Figure 2** and numbered with the corresponding cropped vesicles. (E) CryoEM micrographs of SEC purified vesicles. Cropped images indicate representative purified vesicle particles. Scale bars are 100 nm. Uncropped images are available in **Supplementary Figure 2** and numbered with the corresponding cropped vesicles.

• Related to Fig. 1, the data in supplementary Fig 2 B provided a nice level of precision and could be considered for the primary Fig. 1.

We brought the data from **Supplementary Figure 2B** into the manuscript as **Figure 2B** (see one point above). We added the following text to explain and contextualize the data:

Nanoparticle Tracking Analysis (NTA), an orthogonal method for sizing and quantitating nanoparticles in solution, revealed an average purified vesicle diameter of 118.5 ± 0.7 nm, corroborating the approximate size range of vesicles measured with DLS (Fig. 2B).

• Page 6, line 5. “The 20 nm peak represents small cell-derived particles, including assembled 20 nm E. coli ribosomes62, which we confirmed to be active in our extracts (Supplementary Fig. 1).” It is unclear what is being tested here – is this an isolated fraction containing only the 20 nm peak? The generation of GFP from a crude lysate that contains 20 nm diameter particles is likely the result of ribosomes, but I’m not convinced this experiment demonstrates a direct link.

This demonstration could be excluded or should be re-written to more clearly present the data and rationale.

To clarify our results and rationale, we have reworded the sentence to read:

E. coli ribosomes, which are present at $\sim 1 \mu\text{M}$ in typical CFE reactions, and enabled the production of sfGFP in our CFE reactions (**Supplementary Fig. 1**), are $\sim 20 \text{ nm}$ in size and likely contribute considerably to the signal measured^{73,74}.

While discussed in the next section, the method of cell lysis for the lysate evaluated in Figure 1 should be mentioned ahead of presentation of the data.

We agree with the reviewer. We now mention the previous method of cell lysis within the first few sentences of the results section:

Initially, we used several nanocharacterization techniques to analyze the size of vesicles and visualize these particles in CFE extracts prepared using homogenization and 30,000 x g clarification as described previously³⁴.

• Figure 3. Labelling of all western blots with S12 and S30 headings would be clearer.

We have added these labels with S12 and S30 heading to the Western blots. See below.

• Figure 4A. Semi-quantitative Western blotting should be mentioned in the legend of Figure 4 as the method used to measure glycosylated protein.

We would like to clarify that sfGFP fluorescence was measured and correlated to ^{14}C -Leucine-determined yields to determine the total sfGFP concentrations presented in the original **Figure 4**. The blots were then used to determine the fraction of glycosylated protein. To better describe our methods, we added the following text to the legend of this figure (which is the new **Figure 5**)

For panels A-E a standard curve correlating protein yields derived from ¹⁴C-Leucine counting and sfGFP fluorescence was used to measure total protein concentrations. Quantitative Western blotting was used to measure fraction of glycosylated protein.

During the review process, we also validated improvements in glycoprotein titer using 1 native glycoprotein and 2 conjugate vaccine carrier proteins in **Figure 5F**. We added similar text in the caption to describe these experiments:

For panel F, protein concentrations were measured using ¹⁴C-Leucine incorporation. Fraction of glycosylated protein was measured using autoradiography.

- The method for glycoprotein quantification (ug/mL) in Figure 4 is not clear. How were the units of ug/mL calculated from band intensities? Were titrations of known standards for glycosylated and non-glycosylated forms of GFP evaluated using Western blot to calibrate measurements? Antibody binding to glycosylated protein and non-glycosylated protein is also not necessarily equivalent.

We thank the reviewer for the opportunity to clarify our methods. Western blotting was not used to calculate protein concentrations. ¹⁴C-labeled leucine followed by liquid scintillation counting was used to quantitate protein concentrations. We then used a standard curve correlating ¹⁴C-derived protein concentration with sfGFP fluorescence to calculate concentrations.

The standard curve is provided for the reviewer's reference (right). This was used for the following figures: **Figure 5, Supplementary Figure 4 and Supplementary Figure 9E**. To calculate glycoprotein yields, densitometry of acceptor proteins on Western blots or autoradiograms were used, as is now described in detail in the methods.

For clarity, instead of referencing another publication to describe the quantification, our methods section has now been updated to describes our ¹⁴C quantitation methods, as well as our densitometry measurements from Western blots and autoradiograms. See below for the new text added to the manuscript.

Quantification of CFE and CFGpS protein yields

As described previously, the concentration of cell-free-derived sfGFP was determined by measuring in-extract fluorescence and then converting to protein concentration using a standard curve relating sfGFP fluorescence to protein concentration as determined by a [¹⁴C]-leucine incorporation assay³⁴. Briefly, 2 μ L of cell-free reaction product was diluted into 48 μ L of Ambion nanopure water (Invitrogen, USA). The solution was then placed in a Costar 96-well black assay plate (Corning, USA). Fluorescence was measured using a Synergy H1 microplate reader (Biotek, USA). Excitation and emission wavelengths for sfGFP fluorescence were 485 and 528 nm, respectively. This RFU value was then used to calculate the protein concentration.

Yields of all acceptor proteins (other than sfGFP) were assessed directly via the addition of 10 μ M [¹⁴C]-leucine (PerkinElmer) to the CFGpS reaction to yield trichloroacetic acid-

precipitable radioactivity that was measured using scintillation counting. Soluble fractions were isolated after centrifugation at $\geq 12,000 \times g$ for 15 min at 4 °C. Briefly, 6 μL of the soluble fraction of CFGpS reactions run with 20 min CFPS times were mixed with 6 μL 0.5 M KOH and incubated for 20 min at 37°C. 5 μL of treated sample was then soaked into 2 separate filtermats (PerkinElmer Printer Filtermat A 1450-421) and dried under a heat lamp. One filtermat was washed three times using 5% trichloroacetic acid (TCA) with 15 min incubations at 4°C, and then once with ethanol with a 10 min incubation at room temperature. Following melting of scintillation wax (PerkinElmer MeltiLex A 1450-441) on top of both TCA-precipitated and non-TCA-precipitated filtermats, incorporated radioactivity was measured by a Microbeta2 (PerkinElmer) scintillation counter. Low levels of background radioactivity in S12 and S30 extracts were measured in CFGpS reactions containing no plasmid DNA template and subtracted before calculation of protein yields. The fraction of incorporated leucine (washed/unwashed) was used to determine the amount of protein produced in each reaction.

In order to confirm that results obtained via quantitative Western blotting for quantifying glycoprotein fraction holds across an orthogonal method (which does not require antibody binding), we ran an autoradiogram testing CFGpS in S12 vs. S30 with the sfGFP acceptor as conducted in the manuscript. For reference, these samples are equivalent to the 30 min CFPS time presented in the main body. We indeed observed that trends between ratios of glycosylated and aglycosylated product are consistent with Western blotting. Please see the autoradiogram to the right. Additionally, our experiments with the other non-model proteins (in which glycoprotein was quantitated solely by autoradiogram) show similar improvements with S12. The new methods text is described below.

Autoradiograms of CFGpS reaction products

For sfGFP-based glycosylation experiments, Western blotting of the acceptor proteins followed by densitometry analysis was used to quantitate the fraction of acceptor protein glycosylated (see above for detailed description). For other acceptor proteins, autoradiograms were used to quantitate glycoprotein from CFGpS reaction products using densitometry. Autoradiograms were run by first running SDS-PAGE gels of the soluble fractions of CFGpS reactions (from the same reactions used to calculate yields) using NuPAGE 4-12% Bis-Tris protein gels with MOPS-SDS buffer (Thermo Fisher Scientific, USA). The gels were then dried overnight between cellophane films and then exposed for 48-72 hours to a Storage Phosphor Screen (GE Healthcare). The Phosphor Screen was imaged using a Typhoon FLA7000 imager (GE Healthcare). Autoradiogram gel images were acquired using Typhoon FLA 7000 Control Software Version 1.2 Build 1.2.1.93. Autoradiogram analysis was performed using ImageJ (Version 2.1.0/1.53c, Build 5f23140693) gel analyzer to determine ratios of glycosylated and aglycosylated full-length acceptor protein. Glycoprotein yields were determined by multiplying fraction

glycosylated as determined by ImageJ analysis, by the yields determined from scintillation counting for each replicate.

- The purpose of the insets in Fig. 4B,D containing the star symbols is not clear as it is currently presented. Could this information just be placed in the text? These insets currently look like legends for the data, which is confusing.

Thank you for this helpful comment. We removed the figure insets in the new **Figure 5B, D** and placed this information in the text.

- Overall, excellent work and presentation of an exciting new aspect of cell-free systems!

Thank you so much for your support of publication. We are very grateful.

Reviewers' Comments:

Reviewer #1:

Remarks to the Author:

In the course of the review process, the manuscript was edited and repeatedly revised. As a result, the quality of the manuscript has improved substantially. The revised manuscript is now suitable for publication.

Reviewer #2:

Remarks to the Author:

Dear authors,

It is very helpful to have the Figure 1 schematic included. Abbreviations in the figure (OST, LLO) would benefit from being defined in the figure legend.

The additions/modifications to data establishing membrane vesicles in the crude CFE improve the manuscript.

The clarification to the methods used and the addition of method descriptions for data in Figure 5 is helpful. It is a somewhat complex, but really interesting data set.

It was quite useful to see the standard curve in the response to comments and I would recommend including this in the SI section, but I leave this up to the authors.

The abbreviations "g1" and "g0" in panels 5C and 5E would benefit from being defined in the Figure 5 legend.

REVIEWERS' COMMENTS

Reviewer #1 (Remarks to the Author):

In the course of the review process, the manuscript was edited and repeatedly revised. As a result, the quality of the manuscript has improved substantially. The revised manuscript is now suitable for publication.

We thank the reviewer for their support of the changes made to the manuscript during the review process and appreciate the recommendation for publication.

Reviewer #2 (Remarks to the Author):

Dear authors,

It is very helpful to have the Figure 1 schematic included. Abbreviations in the figure (OST, LLO) would benefit from being defined in the figure legend.

We thank the reviewers for this suggestion and have now defined the abbreviations OST and LLO in the Figure 1 legend. This sentence now reads: "By increasing concentrations of vesicles and membrane-bound glycosylation machinery, oligosaccharyltransferases (OSTs) and lipid-linked oligosaccharides (LLOs), we overcome limitations in cell-free glycoprotein synthesis and increase glycoprotein titers."

The additions/modifications to data establishing membrane vesicles in the crude CFE improve the manuscript.

We thank the reviewer for the support of the changes made to the manuscript during revision and agree that the suggested modifications improve our manuscript.

The clarification to the methods used and the addition of method descriptions for data in Figure 5 is helpful. It is a somewhat complex, but really interesting data set.

We are glad that the reviewer finds the clarifications helpful and finds the additional data an interesting addition to the manuscript.

It was quite useful to see the standard curve in the response to comments and I would recommend including this in the SI section, but I leave this up to the authors.

We thank the reviewer for this suggestion. We have now added the standard curve and associated raw data to the source data file that is provided with the manuscript.

The abbreviations "g1" and "g0" in panels 5C and 5E would benefit from being defined in the Figure 5 legend.

We thank the reviewer for this suggestion and have added the following text to define g1 and g0 in Figure 5c,e captions: "Anti-His and anti-glycan Western blots of acceptor proteins from representative reactions in (B) show glycosylated (g1) and aglycosylated (g0) protein." and "Anti-His and anti-glycan Western blots of acceptor proteins from representative reactions in (D) show glycosylated (g1) and aglycosylated (g0) protein." respectively.